



# Atmospheric Measurements at the Foot and the Summit of Mt. Tai - Part II: HONO Budget and Radical (RO$_x$ + NO$_3$) Chemistry in the Lower Boundary Layer

Chaoyang Xue[1, 2*], Can Ye[1, 6], Jörg Kleffmann[5], Wenjin Zhang[1], Xiaowei He[1, 9], Pengfei Liu[1, 3], Chenglong Zhang[1, 3], Xiaoxi Zhao[1, 9], Chengtang Liu[1, 3], Zhuobiao Ma[1], Junfeng Liu[1, 3], Jinhe Wang[7], Keding Lu[6], Valéry Catoire[2], Abdelwahid Mellouki[4, 8], Yujing Mu[1, 3*]

[1] Research Centre for Eco-Environmental Sciences, Chinese Academy of Sciences, Beijing 100085, China

[2] Laboratoire de Physique et Chimie de l'Environnement et de l'Espace (LPC2E), CNRS–Université Orléans–CNES, Cedex 2, Orléans 45071, France

[3] Centre for Excellence in Regional Atmospheric Environment, Institute of Urban Environment, Chinese Academy of Sciences, Xiamen 361021, China

[4] Institut de Combustion Aérothermique, Réactivité et Environnement, Centre National de la Recherche Scientifique (ICARE-CNRS), Cedex 2, Orléans 45071, France

[5] Physical and Theoretical Chemistry, University of Wuppertal, Gaußstrasse 20, 42119 Wuppertal, Germany

[6] State Key Joint Laboratory of Environment Simulation and Pollution Control, College of Environmental Sciences and Engineering, Peking University, Beijing, 100871, China

[7] School of Municipal and Environmental Engineering, Co-Innovation Centre for Green Building of Shandong Province, Shandong Jianzhu University, Jinan 250101, China

[8] Environmental Research Institute, Shandong University, Qingdao, Shandong 266237, China

[9] University of Chinese Academy of Sciences, Beijing 100049, China

*Correspondence to*:

Chaoyang Xue (chaoyang.xue@cnrs-orleans.fr; 86chaoyang.xue@gmail.com)

Yujing Mu (yjmu@rcees.ac.cn)



**Abstract**

In the summer of 2018, a comprehensive field campaign, with measurements on HONO and related parameters, was conducted at the foot (150 m a.s.l.) and the summit of Mt. Tai (1534 m a.s.l.) in the central North China Plain (NCP). With the implementation of a 0-D box model, the HONO budget with six additional sources and its role in radical chemistry at the foot station were explored. We found that the model default source, $NO + OH$, could only reproduce the observed HONO by 13%, leading to a strong unknown source strength up to 3 ppbv h$^{-1}$. Among the additional sources, the $NO_2$ uptake on the ground surface dominated (~70%) night-time HONO formation, and its photo-enhanced reaction dominated (~80%) daytime HONO formation. Their contributions were sensitive to the mixing layer height (MLH) used for the parameterizations, highlighting the importance of a reasonable MLH for exploring ground-level HONO formation in 0-D models and the necessity of gradient measurements. A HONO/NO$_x$ ratio of 0.7% for the direct emission was inferred and a new method to quantify its contribution to the observations was proposed and discussed. Aerosol-derived sources, including the $NO_2$ uptake on the aerosol surface and the particulate nitrate photolysis, did not lead to significant HONO formation, with their contributions lower than $NO + OH$. HONO photolysis in the early morning initialized the daytime photochemistry at both the foot and the summit stations and also was a substantial radical source throughout the daytime, with contributions higher than or about one-quarter of $O_3$ photolysis to OH initiation at the foot and the summit stations, respectively. Moreover, we found that OH dominated the atmospheric oxidizing capacity in the daytime, while $NO_3$ appeared to be significant at night. Peaks of $NO_3$ time series and diurnal variation reached 22 and 9 pptv, respectively. $NO_3$ induced reactions contribute 18% of nitrate formation potential ($P(HNO_3)$) and 11% of the isoprene ($C_5H_8$) oxidation throughout the whole day. At night, $NO_3$ chemistry led to 51% or 44% of $P(HNO_3)$ or the $C_5H_8$ oxidation, respectively. $NO_3$ chemistry may significantly affect night-time secondary organic and inorganic aerosol formation in this high-$O_3$ region, implying that $NO_3$ chemistry could significantly affect night-time secondary organic and inorganic aerosol formation in this high-$O_3$ region. Considering the severe $O_3$ pollution in the NCP and the very limited $NO_3$ measurements, we suggest that besides direct measurements of $HO_x$ and primary $HO_x$ precursors ($O_3$, HONO, alkenes, etc.), $NO_3$ measurements should be conducted to understand the atmospheric oxidizing capacity and air pollution formation in this and similar regions.

## 1 Introduction

Numerous field campaigns coupled with model simulations have been conducted worldwide to understand the summertime atmospheric chemistry as it is linked to the regional air quality and global climate (Alicke et al., 2003; Elshorbany et al., 2012; Heard et al., 2004; Kanaya et al., 2009, 2013; Michoud et al., 2012; Ren et al., 2003; Rohrer and Berresheim, 2006; Tan et al., 2017; Travis et al., 2020). One of the key issues is the level and the production/loss paths of the atmospheric oxidizing capacity governed by radicals (OH, $NO_3$, etc.). This is also essential in converting primary to secondary pollutants and the removal of greenhouse gases (Lu et al., 2019; Seinfeld and Pandis, 2016).



On a global scale, OH controls atmospheric oxidation. As the detergent in the troposphere, OH can oxidize most trace gases, including inorganic ($SO_2$, $NO_2$, etc.) and organic compounds (VOCs, etc.), and determines the lifetime of greenhouse gases (e.g., $CH_4$). Besides the fast conversion of $HO_2$ to OH (R-1) as part of the radical propagation cycle, primary OH (radical initiation) mainly originates from photolysis reactions, including $O_3$ ((R-2) to (R-4)), HONO (R-5), HCHO ((R-6) to (R-9), and (R-1)), and $H_2O_2$ (R-10), and the ozonolysis of alkenes (not shown in detail here). In particular, HONO photolysis is reported to be an important or even the major OH source in the lower atmosphere of polluted regions, with a contribution of $20 - 90\%$ (Alicke et al., 2003; Elshorbany et al., 2009; Kleffmann et al., 2005; Platt et al., 1980; Xue et al., 2020). However, this process still needs more global quantifications due to the incomplete understanding of HONO formation and its vertical distribution in the atmosphere (Kleffmann, 2007). A state-of-art summary of the reported HONO sources can be found in our recent study (Xue et al., 2020).

$$HO_2 + NO \rightarrow OH + NO_2 \tag{R-1}$$

$$O_3 + hv \rightarrow O(^1D) + O_2 \quad (\lambda < 320 \text{ nm}) \tag{R-2}$$

$$O(^1D) + H_2O \rightarrow 2OH \tag{R-3}$$

$$O(^1D) + M \rightarrow O + M \quad (M=N_2 \text{ or } O_2) \tag{R-4}$$

$$HONO + hv \rightarrow NO + OH \quad (\lambda < 400 \text{ nm}) \tag{R-5}$$

$$HCHO + hv \rightarrow H + HCO \quad (\lambda < 340 \text{ nm}) \tag{R-6}$$

$$HCHO + hv \rightarrow H_2 + CO \quad (\lambda < 365 \text{ nm}) \tag{R-7}$$

$$H + O_2 \rightarrow HO_2 \tag{R-8}$$

$$HCO + O_2 \rightarrow CO + HO_2 \tag{R-9}$$

$$H_2O_2 + hv \rightarrow 2OH \quad (\lambda < 362 \text{ nm}) \tag{R-10}$$

Besides, other oxidants can also be of importance on a regional scale. For example, $NO_3$ radical could be a major oxidant in forests (vegetation shadows slow down its photolysis) or in the nocturnal boundary layer at high $O_3$ regions (Brown and Stutz, 2012). Formed by the reaction of $NO_2 + O_3$ (R-11), high $NO_3$ levels usually occur at night, concerning its very rapid photolysis during the daytime (R-12) and (R-13). Moreover, high $NO_3$ concentrations are only observed for high $O_3$ and medium $NO_x$ concentrations in the absence of significant levels of NO caused by reaction (R-15). Like OH, $NO_3$ also has high reactivity with various trace gases (Brown and Stutz, 2012; Mellouki et al., 2021). For example, $NO_3$ reacts with $NO_2$ to form $N_2O_5$ (R-14), which can undergo hydrolysis on wet surfaces or clouds to produce $HNO_3$ (or $NO_3^-$) (R-16) or decomposition back to $NO_3 + NO_2$ (R-17). $NO_3$ can also react with various organic compounds to form secondary organic aerosol (SOA). For instance, $NO_3$ reacts with isoprene ($C_5H_8$), leading to significant organic nitrates (e.g., alkyl nitrates) production (Rollins et al., 2009).

$$NO_2 + O_3 \rightarrow NO_3 + O_2 \tag{R-11}$$

$$NO_3 + hv \rightarrow NO + O_2 \quad (\lambda < 690 \text{ nm}) \tag{R-12}$$



$$NO_3 + hv \rightarrow NO_2 + O \qquad (\lambda < 690 \text{ nm}) \tag{R-13}$$

$$NO_2 + NO_3 \rightarrow N_2O_5 \tag{R-14}$$

$$NO + NO_3 \rightarrow 2NO_2 \tag{R-15}$$

$$N_2O_5 + H_2O \xrightarrow{surface} 2HNO_3 \ (or \ 2NO_3^-) \tag{R-16}$$

$$N_2O_5 \rightarrow NO_2 + NO_3 \tag{R-17}$$

In the past decade, particle pollution, such as $PM_{2.5}$, is going down while $O_3$ pollution is increasing in many cities of China (Han et al., 2020; Li et al., 2019; Sun et al., 2016, 2019), especially in the North China Plain (NCP), where there exists a large population (>330 million) and air pollution in this region becomes a major environmental risk for public health. This raises effort in exploring the $NO_x$-VOCs-$O_3$ chemistry. Meanwhile, high $O_3$ indicates an enhanced atmospheric oxidizing capacity; that is, elevated OH and $NO_3$ levels are expected. However, OH and $NO_3$ levels, as well as their production (e.g., HONO

photolysis or $NO_2 + O_3$) or loss (e.g., to oxidize primary pollutants) in the high-$O_3$ region of the NCP, by far, are very few reported (Lu et al., 2019; Suhail et al., 2019). Herein, we provided the first HONO measurements at the foot of Mt. Tai (in Tai'an city, a typical urban site), followed by measurements at the summit of Mt. Tai in the summer of 2018. Data from the summit station was presented in the companion paper, in which daytime HONO formation and its role in the atmospheric oxidizing capacity at the summit level were studied. In this paper, coupled with the box model, the HONO budget and the

radical chemistry at the ground level were explored and discussed.

## 2 Methods

### 2.1 Field Campaign

#### 2.1.1 Measurement Site

In the summer (from late May to July) of 2018, a comprehensive field campaign was conducted to understand the atmospheric

oxidizing capacity and $O_3$ pollution in Tai'an, a city in the middle of the NCP. Measurements were conducted both at the ground level (the foot of Mt. Tai, 150 m a.s.l.) and the summit level (the summit of Mt. Tai, 1534 m a.s.l., 36.23°N, 117.11°E). The foot station was inside Shandong College of Electric Power (36.18°N, 117.11°E), which represents a typical urban site. Inside the campus (about 50 ha) frequent traffic was not observed, but it sometimes occurred on the urban roads nearby. Tai'an city has a population of about 5.6 million and is about 60 km south of Jinan city (the capital city of Shandong province,

population: ~8.7 million). Mt. Tai locates in the north part of Tai'an city. Locations of these two stations on the map could be found in the companion ACP paper (entitled "Atmospheric Measurements at the Foot and the Summit of Mt. Tai - Part I: HONO Formation and Its Role in the Oxidizing Capacity of the Upper Boundary Layer").



### 2.1.2 Instrumentation

HONO mixing ratios were continuously measured by the LOPAP technique (LOPAP-03, QUMA GmbH, Germany) (Heland
et al., 2001; Kleffmann et al., 2006) from 29th May to 8th July 2017 at the foot station, followed by measurements at the summit
station from 9th to 31st July 2017. At the foot station, NO-NO$_2$-NO$_x$, O$_3$, CO, and SO$_2$ were online measured by a series of
Thermo Fisher Scientific instruments (42i, 49i, 48i, and 43i, respectively). Because chemiluminescence techniques were
reported to overestimate the NO$_2$ level caused by other NO$_y$ interference, we furtherly corrected the measured NO$_2$ with a
family constraint in a model run (see Section 3.1.2). VOCs (56 species) and OVOCs (15 species) were measured by a
homemade GC-FID instrument (Liu et al., 2016) and the USEPA DNPH-HPLC method (Wang et al., 2020), respectively. Gas-
phase H$_2$O$_2$ was measured by a monitor based on the wet chemical method (AL2021, Aerolaser GmbH, Germany), and details
about the used instrument can be found in Ye et al. (2018). Water-soluble ions (i.e., NO$_3^-$, SO$_4^{2-}$, Cl$^-$, Na$^+$, K$^+$, Ca$^{2+}$, etc.) of
PM$_{2.5}$ were collected on Teflon filters every two hours at a sampling flow of 100 L min$^{-1}$ and analyzed by an ion chromatograph
(Liu et al., 2020).
Meteorological parameters, including atmospheric temperature (T), pressure (p), relative humidity (RH), wind direction (WD),
wind speed (WS), and solar irradiance (Ra) were continuously measured by an auto meteorological station. J(NO$_2$) was
measured by a 4-π filter radiometer (Metcon GmbH, Germany). 10-min and hourly-average data (except for PM$_{2.5}$) were used
for the following analysis (time series and static description) and model simulations, respectively. PM$_{2.5}$ measurement was
obtained from the Tai'an monitoring station (200 m east of the foot station), and only hourly-average data was available. Other
J-values used in this study, including J(HONO), J(O($^1$D)), J(H$_2$O$_2$), J(HCHO)$_{rad}$, and J(HNO$_3$), are calculated by the box model
based on trigonometric SZA function (MCM default photolysis frequency calculation, see Jenkin et al. (1997)) and scaled by
the measured J(NO$_2$). For instance, J(HONO) = J(HONO)$_{model}$*J(NO$_2$)$_{measured}$/J(NO$_2$)$_{model}$.

### 2.2 Model Description

#### 2.2.1 Box Model and Constraints

The **F**ramework for **0**-D **A**tmospheric **M**odeling, F0AM v4.0 (available at https://github.com/AirChem/F0AM) developed by
Wolfe et al. (2016) was used to explore the HONO budget and the radical chemistry. The used chemical mechanism was MCM
v3.3.1, which could be obtained from http://mcm.leeds.ac.uk/MCMv3.3.1/home.htt. Note that the present F0AM model could
also be run with family constraints (see details in Wolfe et al. (2016)), such as the NO$_y$ family, Cl$_y$ family, etc., and hence it
allows us to correct for interferences of the NO$_2$ measurement by the chemiluminescence method (see Section 3.1.2).
The model was constrained by the measured J(NO$_2$), T, RH, P, VOCs, OVOCs, and all the other measured inorganic species,
including the corrected NO$_2$ by the family constraint. Continuous VOCs measurement was available from 12th June to 6th July,
and hence box model simulations were performed during this period. While J(NO$_2$) measurement was available from 16th June,
J(NO$_2$) from 12th to 16th June was estimated through the high quadratic correlation (R$^2$ = 0.96, Figure S1) between J(NO$_2$) and
solar irradiance.



### 2.2.2 Model Scenarios

Table 1 shows the description of different model scenarios. A base case (Sce-0) with all the measured parameters as constraints was run to simulate radicals' concentrations and their production/loss rates. The family constraint was used in this scenario to correct for interferences of $NO_2$ measurements (Section 3.1.2). Meanwhile, the role of HONO in radical chemistry was also explored by several model sensitivity tests with reducing or increasing the constrained HONO.

With the simulated OH and the corrected $NO_2$ from the base case, we could further explore the HONO budget. Three model scenarios were conducted to assess the potential contributions of different HONO sources, including one with only the default model source (Sce-1), and one with all the six additional sources, including direct emission, the dark and the photo-enhanced $NO_2$ uptake on the aerosol and ground surfaces and nitrate photolysis (Sce-2). In Sce-3, photo-enhanced $NO_2$ uptake on the ground surface was reduced by a factor of 10, aerosol-derived sources ($NO_2$ uptake on the aerosol surface or particulate nitrate photolysis) were significantly enhanced to test whether the aerosol-derived sources could well explain the observations.

**Table 1: Description of different model scenarios.**

| Scenarios | Constraints | Objectives |
|---|---|---|
| Sce-0 | All measurements; $NO_z$ family constraint | $NO_2$ correction; radical concentration and chemistry |
| Sce-1 | All measurements + corrected $NO_2$ and simulated radicals from Sce-0 | HONO simulation with NO+OH |
| Sce-2 | Same as Sce-1 | HONO simulation with additional sources |
| Sce-3 | Same as Sce-1, but with reduced ground $NO_2$ uptake and enhanced aerosol-derived sources | Testing the performance of aerosol-derived sources |

## 3. Results and Discussion

### 3.1 Overview and Potential Interferences on the Measurements

#### 3.1.1 Overview of the Measurements

Figure 1 shows the meteorological parameters measured at the foot station. During the campaign, it was generally sunny except slightly rainy (<10 mm) on 9th, 10th, 13th, and 28th and heavy rainy (~100 mm) at night of 25th/26th June. Ambient temperature was normally around 25 °C at night and around 30 °C during the daytime, except for rainy days when the temperature was relatively low. The relative atmospheric humidity (RH) was high (up to 80%) on those rainy or cloudy days and low (around 40%) on other days. Campaign-averaged temperature and RH were 27.5 °C and 46.6%, respectively (Table 2). Air mass observed at this site was originated from multiple directions, including west, south, and east, which can be obtained from the wind rose plot (Figure S2). Wind speed was generally low, with an average of about 2 m s$^{-1}$.





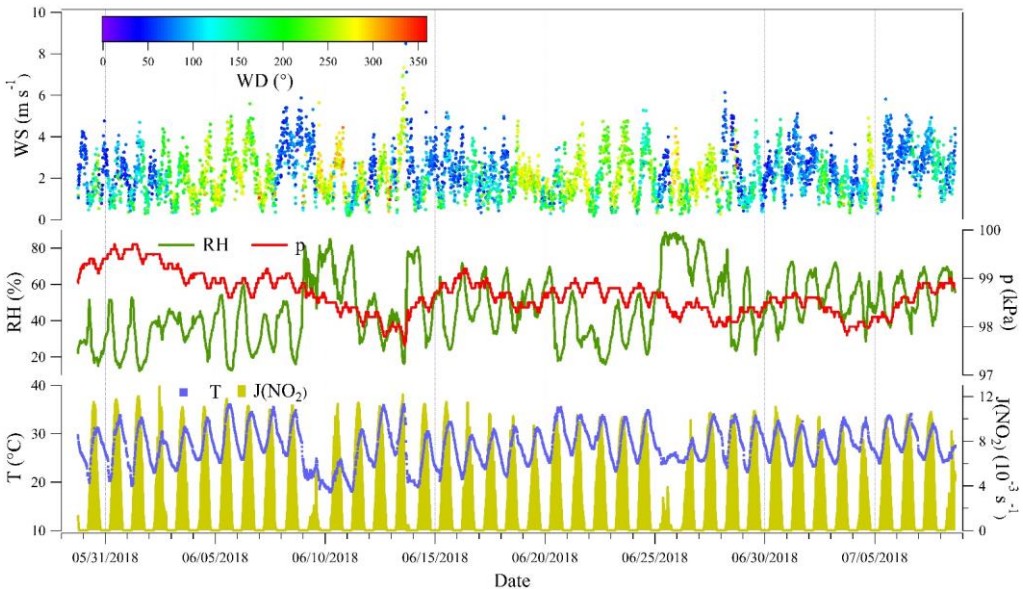

**Figure 1: Meteorological parameters measured at the foot of Mt. Tai during the campaign.**

**Table 2: Statistic summary of meteorological parameters and the measured species. Note that the observation data point number (Obs) of hourly $PM_{2.5}$ is about 1/6 of others (10 min time resolution), and the measured rather than the corrected $NO_2$ was used here. SD: standard deviation; Min: minimum; Max: maximum.**

| Parameters | Mean | SD | Median | Min | Max | Obs |
|---|---|---|---|---|---|---|
| WS (m s$^{-1}$) | 2.2 | 1.1 | 2.1 | 0.2 | 9.7 | 5749 |
| RH (%) | 46.6 | 17.5 | 44.9 | 12.2 | 88.7 | 5749 |
| P (kPa) | 98.7 | 0.4 | 98.6 | 97.6 | 99.7 | 5749 |
| T (°C) | 27.5 | 3.8 | 27.4 | 17.9 | 36.1 | 5749 |
| J($NO_2$) (s$^{-1}$) | 3.2E-03 | 3.7E-03 | 1.0E-03 | 0 | 1.1E-02 | 3183 |
| $O_3$ (ppbv) | 63 | 31 | 62 | 0.1 | 145 | 5727 |
| $PM_{2.5}$ (µg m$^{-3}$) | 29 | 12 | 28 | 10 | 66 | 959 |
| CO (ppmv) | 0.28 | 0.25 | 0.20 | 0.01 | 2.08 | 5717 |
| $SO_2$ (ppbv) | 3.6 | 4.0 | 2.2 | 0 | 36.2 | 5648 |
| NO (ppbv) | 2.0 | 8.3 | 0.3 | 0 | 126.0 | 5749 |
| $NO_2$ (ppbv) | 15.2 | 10.8 | 12.3 | 0 | 78.8 | 5601 |
| HONO (ppbv) | 0.62 | 0.42 | 0.52 | 0.05 | 2.97 | 5423 |

In Figure 2 the measured HONO and related species at the foot station are presented. The measured HONO showed a typical diurnal variation with accumulation after sunset and decay after sunrise. Mixing ratios of the measured HONO varied from 0.05 to about 3 ppbv, with an average of 0.62 ± 0.42 ppbv. The measured $NO_2$ showed a very similar variation to HONO, and their correlation was high (R = 0.73), indicating a potential role of $NO_2$ in HONO formation. Severe $O_3$ pollution (maximum: 145 ppbv) was observed at this site, with $O_3$ levels frequently exceeding the Class-1 limit value (1-h 160 µg m$^{-3}$, equivalent to 82 ppbv at 298K and 101 kPa) of the National Ambient Air Quality Standard of China (GB3095-2012), while the $NO_x$ level





was typically lower than $O_3$. Consequently, a relative low NO was frequently found, whose concentration was generally lower than 1 ppbv, except for some fresh plumes with higher NO concentrations inside. The two primary pollutants, CO and $SO_2$,

were generally lower than 0.5 ppmv and 5 ppbv, respectively, except for several polluted events, within which CO and $SO_2$ reached around 2 ppmv and around 35 ppbv, respectively. However, all the primary pollutants, including NO, CO, and $SO_2$, showed poor correlations with HONO (R = 0.49, 0.44, and 0.13, respectively), implying the minor role of direct emission in HONO formation. The measured hourly $PM_{2.5}$ varied from 10 to 66 µg m$^{-3}$, with an average of 29 µg m$^{-3}$. The correlation of $PM_{2.5}$ and HONO was very low (R = 0.06), suggesting a minor role of aerosol-derived sources in HONO formation. More

discussion on the HONO budget is presented in Section 3.2.

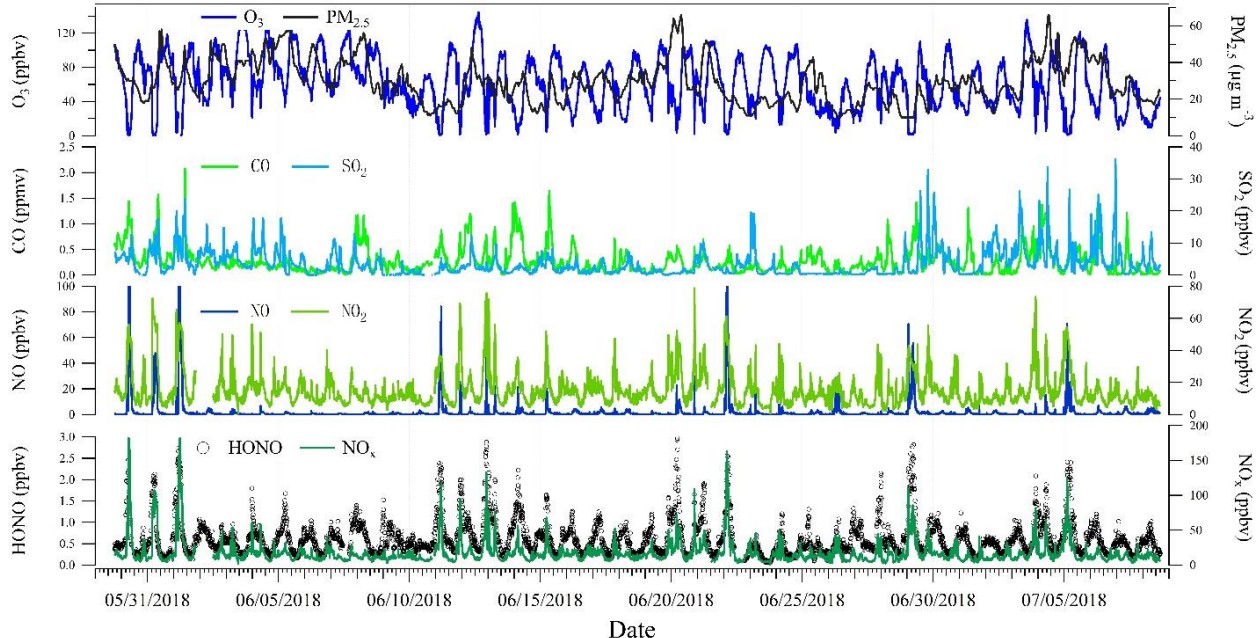

**Figure 2: HONO and related species measured during the campaign.**

Compared to other previous summertime measurements worldwide (Table 3), the measured HONO level at this site is similar to some measurements in China, such as Beijing in 2007 (Hendrick et al., 2014), Beijing in 2008 (Hendrick et al., 2014) and

Guangzhou in 2006 (Yang et al., 2014); in Europe, such as Milan in 1998 (Alicke et al., 2002) and Rome in 2001 (Acker et al., 2006); and in North America, such as New York in 2001 (Ren et al., 2003) and Colorado in 2011 (Vandenboer et al., 2013). Besides, it is lower than measurements in cities during polluted periods, such as Jinan in 2016 (Li et al., 2018), Santiago de Chile in 2005 (Elshorbany et al., 2009), Santiago de Chile in 2009 (Villena et al., 2011), and Mexico in 2003 (Volkamer et al., 2010), but higher than recent measurements near European cities, including Forschungszentrum Karlsruhe (Kleffmann et al.,

2003), Forschungszentrum Jülich (Elshorbany et al., 2012), suburban Paris (Michoud et al., 2014), and Cyprus (Meusel et al., 2016). It is noteworthy that the measured HONO at the foot station is significantly higher than that observed at the summit station in the same summer, indicating possibly different roles and formation paths of HONO at these two stations.





**Table 3: Examples of worldwide HONO measurements at the ground level.**

| Location | Period | Techniques | Mean (pptv) | Range (pptv) | HONO/NO$_x$ % | Reference |
|---|---|---|---|---|---|---|
| **Europe** | | | | | | |
| Milan | May-Jun 1998 | DOAS | 920[a]/140[b] | <4400 | | (Alicke et al., 2002) |
| Pabstthum | Jul-Aug 1998 | DOAS | 330[a]/70[b] | <1200 | 1.4[a]/1.0[b] | (Alicke et al., 2003) |
| Rome | May-Jun 2001 | LP-DOAS, DNPH-HPLC, WEDD | 580 | <2000 | 3[*] | (Acker et al., 2006) |
| Forschungszentrum Karlsruhe | Oct 2001 | LOPAP | 400 | 180-1100 | 1-6[a] | (Kleffmann et al., 2003) |
| Forschungszentrum Jülich | Jun-Jul 2005 | LOPAP | 220 | 50-1100 | 2 (0.6-12) | (Elshorbany et al., 2012) |
| Suburban Paris | Jul 2009 | Ni-troMAC | ~150 | 10-500 | | (Michoud et al., 2014) |
| Cyprus | Jul-Aug 2014 | LOPAP | 35 | <300 | 33 | (Meusel et al., 2016) |
| **South America** | | | | | | |
| Santiago de Chile | Mar 2005 | LOPAP | 2500[a]/2300[b] | 670-7100 | 3.9 (1.3-9.2) | (Elshorbany et al., 2009) |
| Santiago de Chile | Nov 2009 | LOPAP | 1500/1100[**] | 220-3800/150-4600[**] | 2.0 (0.7-5.9) | (Villena et al., 2011) |
| **North America** | | | | | | |
| California | Aug-Sep 1979 | DOAS | 1090[a*]/<280[b] | <4100 | | (Platt et al., 1980) |
| New York | Jul-Aug 2001 | HPLC | 660 | 400-1400 | | (Ren et al., 2003) |
| Colorado | Feb-Mar 2011 | NI-PT-CIMS | 500[a*]/100[b] | <2000 | 3.5-7.6[a] | (Vandenboer et al., 2013) |
| Mexico | Mar-May 2003 | LP-DOAS | 1200[*] | <3000 | | (Volkamer et al., 2010) |
| **China** | | | | | | |
| Beijing | Aug-Sep 2004 | DOAS | | <6100 | 8.4[c] | (Qin et al., 2006) |
| Guangzhou | Jun 2006 | LOPAP | 950[a]/240[b] | 10-5000 | 4.3[a]/4.5[b] | (Li et al., 2012) |
| Yufa | Jul-Aug 2006 | LOPAP | 890[a] /430[b] | 30-3600 | 4.6[a]/4.8[b] | (Yang et al., 2014) |
| Beijing | Aug 2007 | DOAS | 1450 | 440-2900 | 5[c] | (Spataro et al., 2013) |
| Beijing | Jul 2008 | DOAS | 180[d] | 100-800 | 0.8[cd] | (Hendrick et al., 2014) |
| Xianghe | Jun 2012 | DOAS | 90[d] | 100-700 | 1.7[cd] | (Hendrick et al., 2014) |
| Jinan | Jun-Aug 2016 | LOPAP | 1200[a]/1010[b] | <6000 | 6[a]/5[b] | (Li et al., 2018) |
| Tai'an | May-Jul 2018 | LOPAP | 620 | 50-2970 | 4.2 | This study |
| Mt. Tai Summit | Jul 2018 | LOPAP | 133 | 1880 | 6.4 | This study |

[a]: night-time, [b]: daytime, [c]: HONO/NO$_2$, [d]: noontime.

[*]: half of the diurnal maximum.

[**]: 3[rd] and 21[st] floors, respectively.

### 3.1.2 NO$_2$ Interference and Correction

NO$_2$ measured by the chemiluminescence method suffers from the interference of other NO$_y$ species (Villena et al., 2012), primarily including inorganic species such as (measured) HONO, (non-measured) HNO$_3$, HNO$_4$, N$_2$O$_5$, and NO$_3$, peroxyacyl

nitrates (PANs, RC(O)OONO$_2$), organic nitrates (RONO$_2$), and peroxy nitrates (ROONO$_2$), etc. The sum of the latter two was defined here as organic nitrates[*]. Hence, the measured NO$_2$ is the sum of real NO$_2$ and those interference species. HONO was measured and subtracted from the measured NO$_2$, and we defined NO$_2$[*] = the measured NO$_2$ – HONO. As NO$_2$ is the most important HONO precursor, we used the family constraint (NO$_2$[*] = NO$_2$ + HNO$_4$ + 2N$_2$O$_5$ + NO$_3$ + PANs + organic nitrates[*]) in the base case (Sce-0) to separate each species from NO$_2$[*]. In the term of PANs, PAN, PPN, and MPAN (MCM names, see





their structures at http://mcm.leeds.ac.uk/MCMv3.3.1/) were considered. In the class of organic nitrates[*], CH3NO3,
       C2H5NO3, NC3H7NO3, IC3H7NO3, TC4H9NO3, NOA, ISOP34NO3, ISOPANO3, ISOPDNO3, ISOPCNO3, and
       ISOPBNO3 (MCM names) were considered. Considering that $HNO_3$ is very sticky, we expect $HNO_3$ was mostly absorbed by
       the filter and/or sampling tubes before the converter rather than being converted to NO by the converter. Therefore, $HNO_3$ was
       generally not included in the family constraint and only considered for the uncertainty analysis.

Figure 3 shows the model results of the relative contribution of each $NO_2^*$ species to $NO_2^*$. At night with the absence of
       photochemistry, the real $NO_2$ dominated $NO_2^*$ components, with a contribution of >95%, suggesting a small interference on
       the $NO_2$ measurement. However, the contribution of real $NO_2$ was found to decrease during the daytime due to the increasing
       interference. For example, at 11:00, the real $NO_2$ contributed 82% of the $NO_z^*$, which means the interference could be as high
       as +22% (calculated from 18%/82%). In particular, at 11:00, PANs caused the most interference by +21% (calculated from
225    17%/81%).

       The variations of the simulated PANs and $NO_3$ and their ratios to $NO_2$ were similar to previous observations (Brown and Stutz,
       2012; Roberts et al., 1998; Su et al., 2008; Villena et al., 2012; Xue et al., 2011), indicating that the uncertainty of the method
       is small. For the following model simulations and analysis, only the corrected $NO_2$ was used. Besides, Figure S3 exhibits the
       parallel test results, in which $HNO_3$ was included in the family constraint. It can be found that the interference became more
significant; for instance, the interference could be as high as +75% (calculated from 43%/57%, at 11:00). This represents the
       upper limits of the interference if the sampling tubes are heated so that $HNO_3$ could reach the converter.

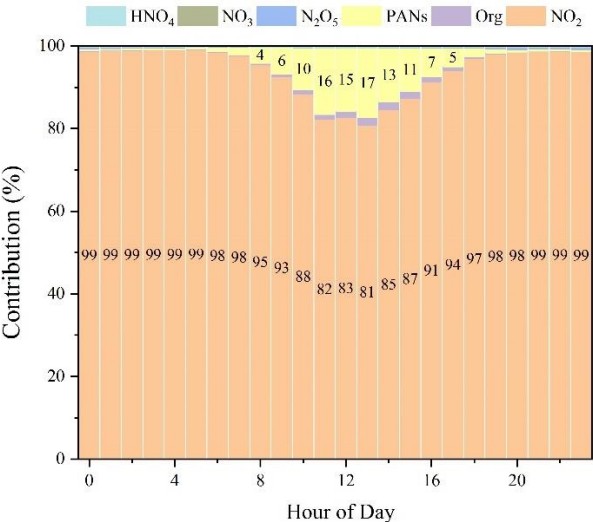

**Figure 3: Relative contribution of each $NO_2^*$ species. PANs = PAN + PPN + MPAN, and Org represents organic nitrates[*] ($RONO_2$ + $ROONO_2$).**

Additionally, as shown in Figure S4, the simulated $HNO_4$ showed 1) a different diurnal variation from, 2) generally $1-2$
       orders of magnitude lower than, and 3) a very poor correlation ($R^2 = 0.06$) with the observed HONO, indicating its negligible





interference on the HONO measurement by the LOPAP technique (Legrand et al., 2014). It is worth noting that for the description of $O_3$ formation in the polluted atmosphere, accurate measurements of VOCs and $NO_x$ are necessary.

## 3.2 HONO Sources and Budget

**3.2.1 Model Default Source (NO + OH) and Unknown Source Strength**

The homogeneous reaction of NO and OH has been adopted as the default HONO source in atmospheric chemistry models, including MCM. Model results from Sce-1 that only contains the homogeneous source with the modeled OH from Sce-0 are shown in Figure 4. Apparently, the source of NO + OH is too small to explain the observed HONO as the simulated one is almost one order of magnitude lower than observations. Its contributions to the measured daytime or night-time HONO are

15% and 12%, respectively.

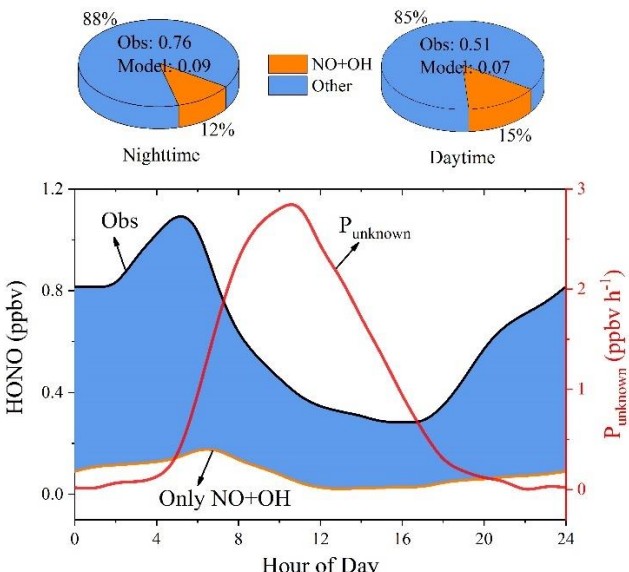

**Figure 4: Simulated HONO by the default mechanism (Sce-1, left axis) compared with the observations (Obs, left axis), unknown source strength ($P_{unknown}$, right axis), and the relative contributions of NO + OH to the observations at night (19:00 – 4:50, left pie chart) and day (5:00 – 18:50, right pie chart), respectively. The shaded area in blue represents the difference between the observation and modeled values.**


Then we calculated the unknown source strength ($P_{unknown}$) based on the following equation (Sörgel et al., 2011; Su et al., 2008).

$$P_{unknown} = \frac{\Delta HONO}{\Delta t} + L(HONO)_{pho} + L(HONO)_{HONO+OH} - P(HONO)_{NO+OH} \quad \text{(Eq-1)}$$

where the HONO loss rates through photolysis ($L(HONO)_{pho}$) and reaction with OH ($L(HONO)_{HONO+OH}$) and production rate

from NO + OH were obtained from the base model scenario (Sce-0 with a constraint of the measured HONO). HONO mixing ratio difference within a one-hour interval, $\frac{\Delta HONO}{\Delta t}$, was calculated by the measurement, and its comparison with $P_{unknown}$ was shown in Figure S5. A typical unknown HONO strength variation, with high values at noontime, was obtained (Figure 4).



$P_{unknown}$ rapidly increased in the morning and peaked nearly 3 ppbv h$^{-1}$ at 11:00, followed by a decrease, revealing a photo-enhanced source. Note that the profile of $P_{unknown}$ was asymmetric around 11:00, indicating the unknown source is not simply
photolytic but also includes its precursors (e.g., $NO_2$). The possible additional HONO sources that are responsible for $P_{unknown}$ are discussed in the following section.

### 3.2.2 Additional Sources vs. $P_{unknown}$

### 3.2.2.1 Direct Emission: $\Delta HONO/\Delta NO_x$ Ratio

The $\Delta HONO/\Delta NO_x$ ratio for the direct emission was determined from fresh plumes, which reached the following requirements:
1) at night when photolysis was absent, 2) rapid NO increase by >10 ppbv within 10 min. Only 17 cases were obtained throughout the campaign due to the persistent high $O_3$ and the fast $NO$-to-$NO_2$ conversion, for which the inferred $\Delta HONO/\Delta NO_x$ might be overestimated. In Table 4 the obtained $\Delta HONO/\Delta NO_x$ was shown, varying from 0.18% to 1.86%, with an average of 0.98% and a median of 0.90%. The inferred value might be larger than the real one as $NO_2$-to-HONO conversion leads to a positive interference, which is consistent with that the inferred $HONO/NO_x$ is generally higher in high
RH conditions (in favor of $NO_2$-to-HONO conversion) (Figure 5). Also, we found that the observed $HONO/NO_x$ is convergent as $NO/NO_2$ increases (Figure 5), which allows a further correction on $\Delta HONO/\Delta NO_x$. The reported $NO/NO_2$ ratios from the combustion process vary from digits to hundreds, e.g., 6.7 in Wuppertal (Kurtenbach et al., 2012), ~18 in Denver (Wild et al., 2017), 5 – 30 in London (Carslaw and Beevers, 2005), and 13 – 43 from China IV/V vehicles (He et al., 2020). Furthermore, in the emission inventory, the $NO/NO_2$ emission ratio in the NCP is about 9 (Zhang et al., 2009). However, the measured
night-time $NO/NO_2$ ratios were less than 3 (Figure 5), much lower than that from on-road measurements, indicating the obtained plumes were not fresh enough. By using a typical $NO/NO_2$ ratio of 10 from car exhaust, the calculated $HONO/NO_x$ through the convergent function is 0.7%, similar to that obtained from laboratory or tunnel experiments (Kirchstetter et al., 1996; Kurtenbach et al., 2001; Liu et al., 2017).

Considering that HONO from direct emission ($HONO_{emi}$) is likely significantly overestimated with a constant $\Delta HONO/\Delta NO_x$
because of different lifetimes of HONO ($\tau(HONO)$) and $NO_x$ ($\tau(NO_x)$) in the daytime (also see Section 3.1.1 where very poor correlations of HONO with primary pollutants were presented and the minor role of direct emission in HONO formation was inferred). Then we calculated $\tau(HONO)$ and $\tau(NO_x)$ (see method in Section 1 of the Supporting Information). As shown in Figure S6A, daytime $\tau(NO_x)$ was typically one order of magnitude longer than $\tau(HONO)$ (Figure S6A), indicating the remarkable overestimation of $HONO_{emi}$ to the measured HONO when using a constant $\Delta HONO/\Delta NO_x$ (Figure S6B). Hence,
$HONO_{emi}$ was quantified by the following equations:

$$HONO_{emi} = 0.7\% \times [NO_x] \text{ (night-time)} \tag{Eq-2}$$

$$HONO_{emi} = 0.7\% \times [NO_x] \times \frac{\tau(HONO)}{\tau(NO_x)} \text{ (daytime)} \tag{Eq-3}$$





In summary, direct emission contributed about 1 – 26% of the measured HONO, with an average of 13%. Moreover, the new method developed here may have uncertainties but largely reduced the significant overestimation of $HONO_{emi}$ to the observations in the daytime compared to using only a constant $\Delta HONO/\Delta NO_x$ (Figure S6B).

**Table 4: $\Delta HONO/\Delta NO_x$ ratios determined from night-time (19:00 – 4:50) data of the campaign.**

| Date | Period | HONO | | NO | | $NO_x$ | | $\Delta HONO/\Delta NO_x$ |
|---|---|---|---|---|---|---|---|---|
| 1st June | 2:20-2:30 | 1.58 | 1.77 | 3.1 | 13.5 | 64.4 | 75.8 | 1.67% |
| | 4:20-4:30 | 2.67 | 2.74 | 0.6 | 38.2 | 36.0 | 74.2 | 0.18% |
| 11th June | 3:00-3:10 | 1.32 | 1.71 | 0.2 | 18.9 | 24.9 | 50.0 | 1.55% |
| | 3:50-4:20 | 1.68 | 2.21 | 15.7 | 71.7 | 51.3 | 107.0 | 0.95% |
| | 4:30-4:40 | 1.91 | 2.31 | 41.2 | 72.4 | 75.3 | 107.3 | 1.25% |
| | 22:00-22:10 | 1.41 | 1.69 | 1.0 | 16.6 | 41.1 | 78.4 | 0.75% |
| | 23:30-23:40 | 1.74 | 1.99 | 1.7 | 17.0 | 53.2 | 71.1 | 1.40% |
| 12th June | 22:10-22:30 | 2.64 | 2.78 | 0.8 | 59.0 | 68.7 | 132.9 | 0.22% |
| 14th June | 3:00-3:10 | 1.43 | 1.76 | 6.6 | 21.2 | 38.8 | 56.5 | 1.86% |
| 20th June | 20:50-21:10 | 0.94 | 1.65 | 0.9 | 29.7 | 30.0 | 108.5 | 0.90% |
| 22nd June | 2:10-2:20 | 1.29 | 1.98 | 4.6 | 95.1 | 51.8 | 147.2 | 0.72% |
| | 2:40-3:00 | 1.58 | 2.20 | 38.3 | 120.1 | 83.1 | 163.7 | 0.77% |
| 29th June | 1:00-1:20 | 0.96 | 2.26 | 3.1 | 70.5 | 24.3 | 109.4 | 1.53% |
| | 2:20-2:40 | 0.34 | 0.40 | 7.0 | 21.1 | 45.6 | 59.6 | 0.43% |
| | 4:20-4:30 | 1.79 | 1.97 | 11.7 | 42.2 | 44.0 | 74.0 | 0.60% |
| 5th July | 1:10-1:30 | 0.77 | 1.38 | 0.1 | 18.2 | 33.8 | 70.6 | 1.66% |
| | 2:10-2:30 | 1.09 | 1.26 | 6.4 | 71.0 | 56.2 | 124.6 | 0.25% |
| Mean | | | | | | | | 0.98% |

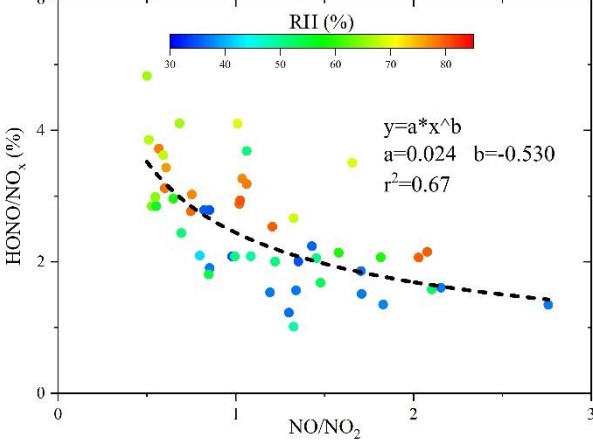

**Figure 5: The inferred direct emission ratio ($HONO/NO_x$) and $NO/NO_2$ colored by RH. Only data with $NO/NO_2 >0.5$ was shown as lower ones indicate much aged rather than fresh plumes.**



### 3.2.2.2 NO₂ Uptake on the Aerosol Surface

Parameterizations of HONO formation from the $NO_2$ uptake on the aerosol surface without $(P(HONO)_{a\_dark})$ and with $(P(HONO)_a)$ photo-enhanced effects are described by (Eq-4) and (Eq-5), respectively. In (Eq-4) and (Eq-5), HONO yields of 50% and 100% were considered for the dark and the photo-enhanced $NO_2$ conversion, respectively (Finlayson-Pitts et al.,

2003; George et al., 2005). A relatively large $NO_2$ uptake coefficient $\gamma_{a\_dark}$ of $1\times10^{-5}$ was used here to represent its upper limit. Its overestimation should not cause significant uncertainties as $P(HONO)_{a\_dark}$ was negligible to HONO formation (see the following discussion). $NO_2$ uptake coefficient $\gamma_a$ values of $1.3\times10^{-4}$ (overestimated one derived from the summit measurement) and $2\times10^{-5}$ (popularly used one derived from laboratory experiments) were used in (Eq-5) to constrain the upper limit and general one of $P(HONO)_a$.

$$P(HONO)_{a\_dark} = \frac{v(NO_2)\times S_a\times[NO_2]}{8} \times \gamma_{a\_dark}, \qquad\qquad \textbf{(Eq-4)}$$

$$P(HONO)_a = \frac{v(NO_2)\times S_a\times[NO_2]}{4} \times [\gamma_a \times \frac{J(NO_2)_{measured}}{0.005\ s^{-1}}], \qquad\qquad \textbf{(Eq-5)}$$

where $\upsilon(NO_2)$, $S_a$, $[NO_2]$, and $J(NO_2)_{measured}$ denote the average $NO_2$ molecular speed (m s$^{-1}$), aerosol surface density (m$^{-1}$), $NO_2$ concentration (ppbv), and the measured $NO_2$ photolysis frequency (s$^{-1}$). As aerosol size distribution measurement was not available at the foot station, we estimated $S_a$ based on the measured $PM_{2.5}$ concentrations because they were highly correlated.

For instance, measurements at the summit station during this campaign and other sites in the NCP found high correlations between $PM_{2.5}$ and $S_a$ (derived from particle size distribution measurement) with a $S_a/PM_{2.5}$ ratio of about $8\times10^{-6} – 1.3\times10^{-5}$ m$^2$ µg$^{-1}$ (Wu et al., 2008; Xue et al., 2020). Here a $S_a/PM_{2.5}$ ratio of $1.0\times10^{-5}$ m$^2$ µg$^{-1}$ was used, and its uncertainty will not cause significant changes in HONO simulation because of its small contribution (see the following discussion).

Diurnal variations of $P(HONO)_{a\_dark}$ and $P(HONO)_a$, in comparison with $P_{unknown}$ and $P(HONO)_{NO+OH}$, are shown in Figure 6A.

Clearly, both $P(HONO)_{a\_dark}$ and $P(HONO)_a$ ($\gamma_a = 2\times10^{-5}$) were negligible compared to daytime $P_{unknown}$. $P(HONO)_a$ increased with $\gamma_a$, but even when using an extremely high $\gamma_a = 1.3\times10^{-4}$, it was still too small to be comparable to $P(HONO)_{NO+OH}$ and far from explaining $P_{unknown}$, revealing minor impacts of $P(HONO)_{a\_dark}$ and $P(HONO)_a$ in HONO formation, particularly during the daytime.





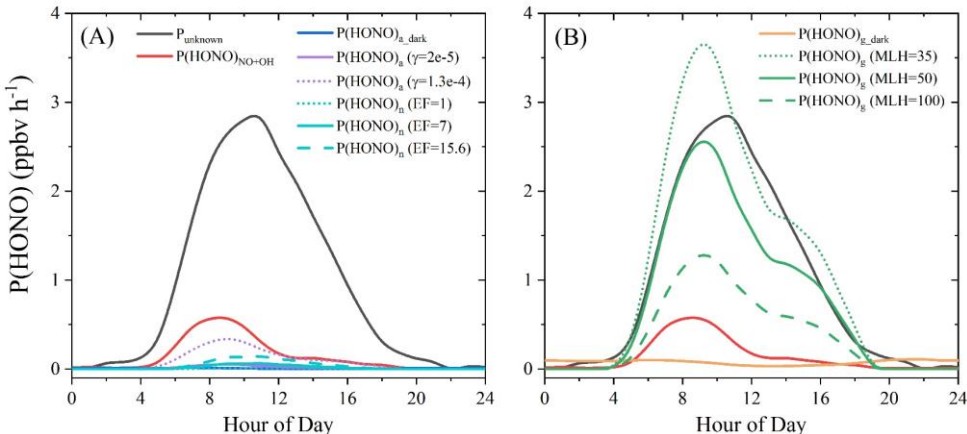

**Figure 6: HONO production rates from different ((A): aerosol-derived, (B): ground-derived) sources and unknown HONO source strength.**

### 3.2.2.3 pNO₃ Photolysis

Parameterization of HONO formation from particulate nitrate photolysis (P(HONO)$_n$) is presented in (Eq-6). Recent studies found that EF values were generally lower than one magnitude, for instance, 7 from a field study (Romer et al. 2018) and ~1

from laboratory studies (Laufs and Kleffmann, 2016; Shi et al., 2021; Wang et al., 2021). Hence EF value of 7 was used in the P(HONO)$_n$ calculation, and values of 1 and 15.6 (overestimated one derived from the summit measurement) were also used to test the sensitivities.

$$P(HONO)_n = pNO_3 * J(HNO_3) * EF, \qquad\qquad\qquad \text{(Eq-6)}$$

where pNO$_3$ and J(HNO$_3$) represent the measured particulate nitrate (with unit converted from µg m$^{-3}$ to ppbv) and the

photolysis frequency of gas-phase HNO$_3$ (s$^{-1}$), respectively.

Diurnal variations of P(HONO)$_n$ with different EF values are shown in Figure 6A. With EF varying from 1 to 7, P(HONO)$_n$ was 1 – 2 orders of magnitude lower than P$_{unknown}$. Even using a high EF = 15.6, P(HONO)$_{nitrate}$ was still significantly less than half of P(HONO)$_{NO+OH}$. Therefore, model results constrained by field measurements and recent kinetics suggested that the two aerosol-derived sources (NO$_2$ conversion and nitrate photolysis) may not have significant impacts on daytime HONO

formation, with their contributions significantly lower than half of P(HONO)$_{NO+OH}$.

### 3.2.2.4 NO₂ Uptake on the Ground Surface

Parameterizations of HONO production from the NO$_2$ uptake on the ground surface without (P(HONO)$_{g\_dark}$) and with (P(HONO)$_g$) photo-enhanced effects are demonstrated in (Eq-7) and (Eq-8), respectively. NO$_2$ uptake coefficients of γ$_{g\_dark}$ and γ$_g$ were set to 1.6×10$^{-6}$ and 2×10$^{-5}$ (Han et al., 2016; Stemmler et al., 2006, 2007), respectively. The photo-enhancement

effect was reflected by $\frac{J(NO_2)_{measured}}{0.005\ s^{-1}}$ (Vogel et al., 2003; Wong et al., 2013; Xue et al., 2020).

$$P(HONO)_{g\_dark} = \frac{v(NO_2) \times [NO_2]}{8 \times \text{MLH}} \times \gamma_{g\_dark}, \qquad\qquad\qquad \text{(Eq-7)}$$



$$P(HONO)_g = \frac{v(NO_2) \times [NO_2]}{4 \times MLH} \times \gamma_g \times \frac{J(NO_2)_{measured}}{0.005 \ s^{-1}}, \qquad \textbf{(Eq-8)}$$

It can be found that one of the most important parameters for calculating ground HONO formation in a box model is the mixing layer height (MLH) as it is part of the denominators in both (Eq-7) and (Eq-8). MLH for HONO should be significantly lower than the boundary layer height (BLH) due to its formation on the ground level and short lifetime, which could be confirmed by the gradient measurements (Kleffmann et al., 2003; Meng et al., 2020; Vogel et al., 2003; Wang et al., 2019; Wong et al., 2012; Xing et al., 2021; Ye et al., 2018b). For instance, a recent gradient HONO measurement by the MAX-DOAS technique in southwest China found a very rapid HONO decrease as increasing altitude from 0 to 4 km (Xing et al., 2021). When considering their measurement at 17:00 (UTC+8) as an example, HONO levels rapidly decreased from 4.8 ppbv at the ground level (~4 m above the ground surface) to 1.6, 0.7, 0.3, 0.2, and 0.1 ppbv averaged in height ranges of 0 – 100, 100 – 200, 200 – 300, 300 – 400, and 400 – 500 m above the ground level, respectively. In contrast, both $NO_2$ and aerosol extinction remarkably increased from the ground level to about 200 m above the ground level and then decreased with altitude (>200 m), indicating that 1) ground-derived sources dominated daytime HONO formation; 2) the MLH for HONO was much less than 100 m, and 3) significant overestimation, i.e., by a factor of >3 in Xing et al. (2021), could be expected if using measurements on the ground surface to represent the average HONO within an MLH higher than 100 m. Therefore, 50 m were used to scale the MLH with sensitivity tests on 35 and 100 m. Similar values (25 – 100 m) were also used in previous box model studies (Lee et al., 2016; Xue et al., 2020, 2021). It should be highlighted that a box model as used in the present study is not an ideal tool for studying a ground source when comparing with near ground surface measurements in the atmosphere. For future, gradient measurements are recommended, which should be compared with 1-D model simulations.

Diurnal variations of $P(HONO)_{g\_dark}$ and $P(HONO)_g$, in comparison with $P_{unknown}$ and $P(HONO)_{NO+OH}$, are shown in Figure 6B. $P(HONO)_{g\_dark}$ was the largest HONO source during the night-time, while it was negligible during the daytime, which is consistent with many previous studies (Li et al., 2010; Liu et al., 2019; Vogel et al., 2003; Xue et al., 2020; Zhang et al., 2019b, 2019a, 2016). With the photo-enhanced effect, $P(HONO)_g$ showed a similar shape and a similar level to daytime $P_{unknown}$, indicating the potential dominance of $P(HONO)_g$ in the daytime HONO formation. When changing MLH to 100 (or 35) m, the level of $P(HONO)_g$ became much lower (or higher) than $P_{unknown}$, for which they were discussed here as sensitivity tests on MLH but not used in Sce-2. Small differences in the shapes of measured and modeled results may be also caused by the variable MLH induced by variable vertical mixing in the atmosphere and the variable photolytic lifetime of HONO during the daytime.

### 3.2.3 HONO Budget

Along with the previous discussion, we conducted a model run (Sce-2) with all the discussed HONO sources. As shown in Figure 7, the model with present HONO source parametrizations showed magnificent performance in predicting HONO as the time series of the modeled HONO was very consistent with those of the observation in both variations and levels, except during the period from 25[th] to 28[th] June (because of heavy rain, see the next section), indicating reasonable parameterizations of the





HONO sources. In particular, the model exhibited very high performance in predicting noontime (10:00 – 16:00) HONO as

the modeled HONO was very close to the observed HONO (Figure 7B). Moreover, in Sce-3 we reduced $\gamma_g$ by a factor of 10 and enlarged $\gamma_a$ from $2\times10^{-5}$ to $1.2\times10^{-3}$ or EF from 7 to 400. We found that the model could also generally predict the observed HONO levels (Figures S7A and S8A) but largely failed to reproduce the noontime observations in levels and variations (Figures S7B and S8B), reinforcing the non-dominated roles of aerosol-derived sources in the daytime HONO formation.

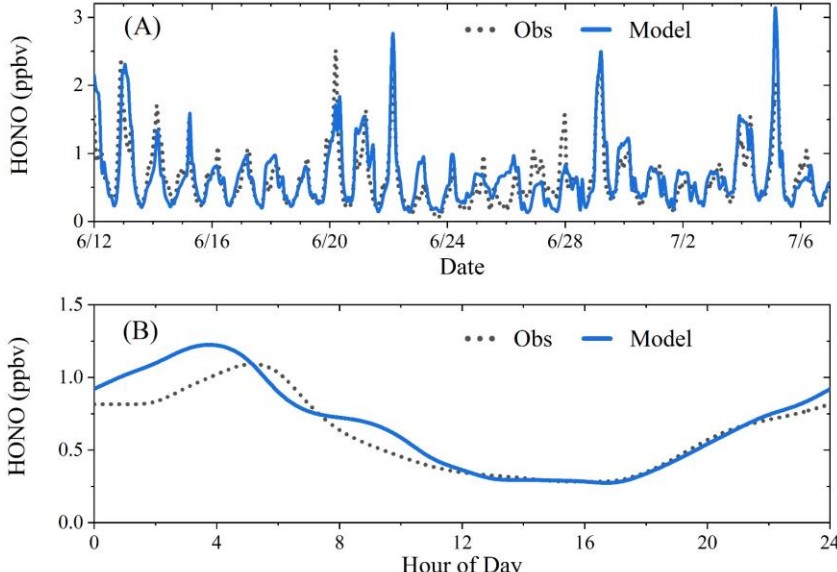

**Figure 7: Modeled HONO mixing ratios (Model, in blue) in comparison with observations (Obs, in black). (A): time series; (B): diurnal variations. Note that HONO_emi was included in the modeled HONO.**

Figure 8 displays the relative contributions of different HONO sources at different hours. It clearly shows that dark $NO_2$ uptake on the ground surface dominated (~70%) night-time HONO formation while photo-enhanced $NO_2$ uptake on the ground surface dominated (~80%) daytime HONO formation. $P(HONO)_{NO+OH}$ played a moderate role throughout the whole day, with

a contribution of 5 – 15% except for a relatively larger contribution (~20%) in the early morning due to high NO levels. Direct emissions made moderate contributions of 15 – 25% at night but negligible ones during daytime. Contributions of $P(HONO)_{a\_dark}$, $P(HONO)_a$, and $P(HONO)_n$ were always lower than 10%, and their contributions could be even smaller when using smaller kinetic parameters derived in recent studies. Therefore, aerosol-derived HONO sources may not significantly contribute to HONO formation at this site (Chen et al., 2019; Neuman et al., 2016; Sarwar et al., 2008; Vogel et al., 2003;

Wong et al., 2013; Xue et al., 2020; Zhang et al., 2016, 2019b).

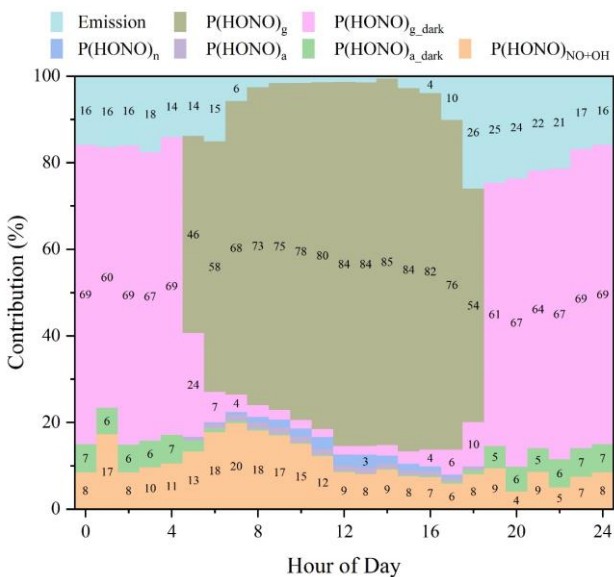

**Figure 8: Relative contributions of different HONO sources. Note that the contribution from direct emission was calculated based on the ratio of HONO$_{emi}$ to the observed HONO.**

### 3.2.4 Other Potential Sources

As discussed before, the model (Sce-2) could generally well reproduce most observations except for the period from 25$^{th}$ to 27$^{th}$ June. A significant overestimation occurred from midday of 25$^{th}$ to the morning of 26$^{th}$, which was caused by the enhanced wet/dry deposition due to the heavy rain (>100 mm, Figure S9) on the night of 24$^{th}$/25$^{th}$. In contrast, from midday of 26$^{th}$ to the night of 27$^{th}$/28$^{th}$, a significant underestimation by the model was obtained. Besides, an elevation of HONO/NO$_x$ was found during this period (Figure S9). This might be caused by 1) the enhanced HONO emission from urban soil or 2) the enhanced

NO$_2$ uptake on the ground surface. The former one may occur through biological processes observed in the laboratory experiments or field measurements over the agricultural fields (Oswald et al., 2013; Scharko et al., 2015; Tang et al., 2019; Xue et al., 2019), while evidence for its occurrence on the urban soil after the rain was still not sufficient. At 13:00 on 26$^{th}$ or 27$^{th}$ June, the model predicted lower HONO by almost a factor of 2 – 4 (observation: 0.45 or 0.45 ppbv; model: 0.13 or 0.21 ppbv), which needs an enhancement of at least 2 – 4 in $\gamma_g$ if using NO$_2$ uptake on the ground surface to explain the

underestimation. Current laboratory experiments have studied the enhancement effect of atmospheric RH (in the range of 10 – 70%) on the NO$_2$ uptake coefficient on the surface of target substances and the enhancement factor was less than 3 (Han et al., 2016; Stemmler et al., 2006, 2007). Campaign averages of the measured NO$_2$ and RH at 13:00 were 7.4 ppbv and 35.5%, respectively. At 13:00 on 26$^{th}$ (or 27$^{th}$) June, the measured NO$_2$ of 7.9 (or 4.3) ppbv was similar to (or lower than) the campaign average, but RH of 67.6% (or 53.1%) was significantly higher than the campaign average but still in the range (10 – 70%)

where RH showed an enhancement (less than 3) effect on $\gamma_g$. Hence, after rain, the enhanced NO$_2$ uptake was likely to be responsible for the underestimation. Meanwhile, soil HONO emission may co-exist but more evidence was needed. However, the impact of direct water addition to those substances (e.g., rainwater on the ground surface) was still not clear. i.e., it may



enhance $NO_2$ uptake and/or soil emission to produce HONO or deposition to consume HONO). Further studies may explore the impact of rain on urban soil surface processes, such as the soil HONO emission flux and $NO_2$ uptake kinetics.

**3.3 Radical Chemistry**

**3.3.1 Role of HONO in Radical Concentrations**

Figure 9 shows the simulated radical concentrations in different model scenarios where their sensitivities to the constrained HONO were tested. It can be obtained that $RO_x$ radicals (OH, $HO_2$, and $RO_2$) were significantly affected by the constrained HONO, implying the vital role of HONO in the $RO_x$ budget. For instance, the peak OH concentration in the base case was 0.42 pptv (equivalent to $1.0\times10^7$ molecules $cm^{-3}$). It decreased to 0.37 (or 0.32) pptv when HONO was reduced by 50% (or 100%) and increased to 0.46 (or 0.51) pptv when HONO was enlarged by 50% (or 100%). In contrast, modeled $NO_3$ concentrations showed very small variations whether HONO was reduced or enlarged, which is because $NO_3$ concentration is mainly governed by the levels of $O_3 + NO_2$ during night-time when HONO has no impact on radical levels caused by the missing photolysis. Nevertheless, the almost same radical concentrations in case NO + OH and case -100% indicate the minor role of NO + OH in the radical budget as this OH sink is exactly compensated by the OH production through (R-5).

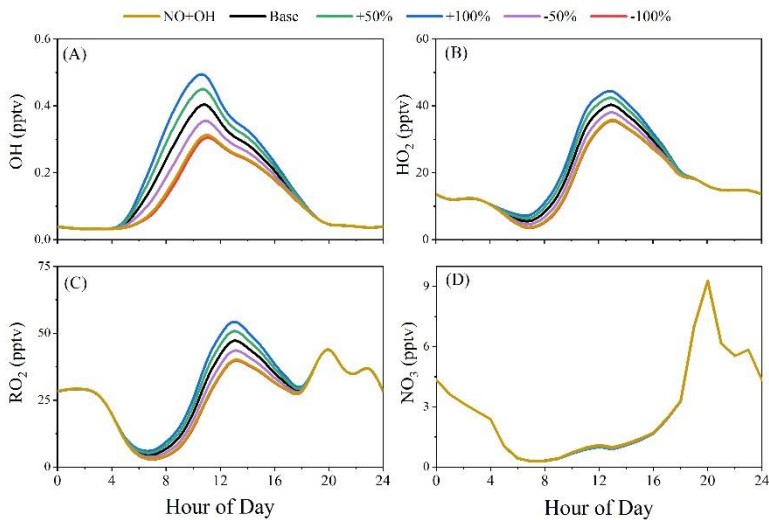

**Figure 9: Simulated concentrations of (A): OH, (B): $HO_2$, (C): $RO_2$, and (D): $NO_3$ in different scenarios: NO + OH: only with the homogeneous source; Base: constrained by the observed HONO; +50%: constrained by the observed HONO×1.5; +100%: constrained by the observed HONO×2; -50%: constrained by the observed HONO×0.5; -100%: constrained by HONO = 0.**

**3.3.2 Radical Production/Loss Rates and Reactivity**

Figure 10A and Figure 10B illustrate the production/loss rates of OH and $NO_3$, respectively. The total production rates of these radicals were similar to their loss rates due to their short lifetimes and high reactivities. For OH (Figure 10A), its largest source was the reaction of $HO_2 + NO$, which is part of the propagation cycle and which is not a radical initiation source (Elshorbany et al., 2010). HONO photolysis was the second-largest OH source, and it is expected to be the largest primary OH source after



subtracting OH loss through HONO + OH and NO + OH (see Section 3.3.4). Reactions with $NO_2$, CO, and $C_5H_8$ acted as the top three OH sinks but did not dominate OH loss due to high OH reactivity caused by various other reactions, particularly those with other VOCs (see below).

Figure 10C and S10A show the OH reactivity with different classes of pollutants and their relative contributions, respectively. Total OH reactivity showed a small peak of 20 $s^{-1}$ in the morning and then kept almost constant around 17 $s^{-1}$. Among different

classes of pollutants, the measured inorganics (including – ordered by OH reactivity contribution – $NO_2 > CO > NO > O_3 >$ $HONO > SO_2 > H_2O_2$) contributed the largest OH reactivity with values in the range of 2.6 – 8.4 $s^{-1}$. Their total contribution was larger in the morning (43%) due to high NO, $NO_2$, and CO levels (Figure 2) and decreased to 15% at noontime. Reactivities with the measured alkanes, alkenes, aromatics, and OVOCs were 0.95 – 1.2 $s^{-1}$, 3.3 – 3.9 $s^{-1}$, 2.2 – 2.9 $s^{-1}$, and 1.65 – 1.9 $s^{-1}$, leading to relative contributions of around 5 – 7%, 18 – 22%, 11 – 17%, and 9 – 12% throughout the whole day, respectively.

Likewise, $C_5H_8$ alone contributed 4% of the OH reactivity in the early morning (0.85 $s^{-1}$), and its contribution increased to 12% at noontime (2.1 $s^{-1}$) as a result of high levels of $C_5H_8$ and OH at noontime.

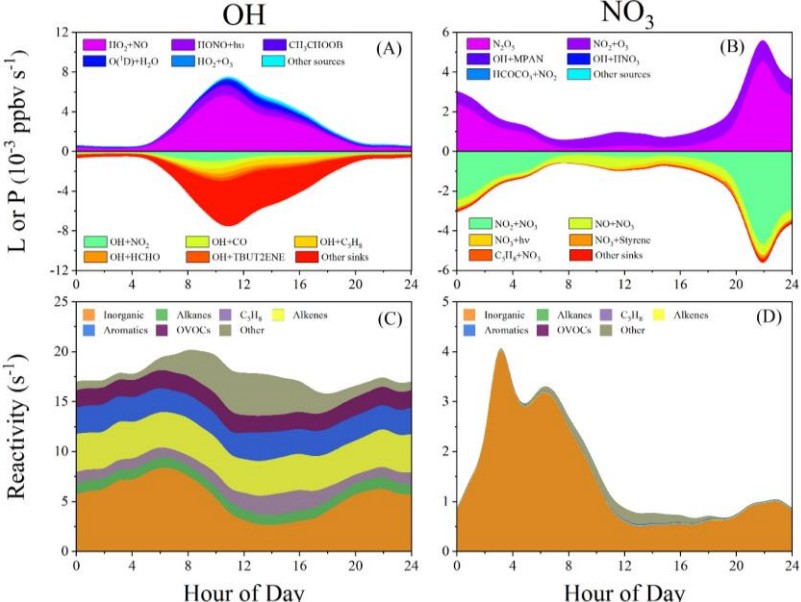

**Figure 10: Production (P) and loss (L) rates of (A): OH and (B): $NO_3$; and reactivities of (C): OH and (D): $NO_3$ with measured species. In A and B, the top five sources or sinks are shown, and all the others are summarized in "Other sources" or "Other**

**sinks". In (C) and (D), reactivities with all the unmeasured species are summarized in "Other".**

Figure 10D and S10B show $NO_3$ reactivity with different pollutant classes and their relative contributions, respectively. Compared with the total OH reactivity, the total $NO_3$ reactivity exhibited lower values and a different variation profile. It showed a minimum of 1 $s^{-1}$ at noontime and increased to around 4 $s^{-1}$ at 2:00. In addition to the $N_2O_5$ decomposition (R-17), $NO_2 + O_3$ (R-11) is the most important $NO_3$ source, which is also, in fact, the most important net $NO_3$ source, considering the

same amount of $NO_3$ loss during $N_2O_5$ production through (R-14). $NO_3$ loss was dominated by photolysis and reactions with





NO during the daytime and reactions with $NO_2$ at night. More discussion on $NO_3$ chemistry is presented in the following section.

### 3.3.3 $NO_3$ Chemistry

As shown in Figure 9D, high $NO_3$ levels (diurnal peak: 9.3 pptv, time-series peak: 22 pptv) were simulated by the model. High

$NO_3$ concentrations, as well as its high reactivity (Figures 10D), generally appeared at night (18:00 to 4:00 in the next day) when OH was very low and $NO_3$ was not lost by photolysis, indicating that the $NO_3$-initialized chemistry may play an important role in night-time chemistry at this site. To verify this implication, we compared the $C_5H_8$ oxidation and nitrate formation through $NO_3$-induced reactions with other paths.

### 3.3.3.1 $C_5H_8$ Oxidation

Figure 11 shows the $C_5H_8$ loss rates ($L(C_5H_8)$) through different oxidation paths and their relative contributions. $L(C_5H_8)$ through $O_3$ was generally in the range of $1.0 – 3.2 \times 10^{-5}$ ppbv s$^{-1}$. $L(C_5H_8)$ through OH showed high values in the daytime and low ones in the night-time. On the contrary to OH, low $L(C_5H_8)$ through $NO_3$ occurred in the daytime and high one occurred in the night-time. On average, $L(C_5H_8)$ through OH, $O_3$, and $NO_3$ oxidation were $3.6 \times 10^{-4}$, $2.0 \times 10^{-5}$, and $4.5 \times 10^{-5}$ ppbv s$^{-1}$, with relative contributions of 84%, 5%, and 11%, respectively. During the daytime, $L(C_5H_8)$ through OH oxidation was

generally one order of magnitude higher than those through $NO_3$ or $O_3$ oxidation, leading to a dominated $C_5H_8$ loss contribution of generally >90% through OH oxidation (Figure 11B). However, at night, OH was much lower and $NO_3$ was higher due to the absence of photochemistry, resulting in an increasing contribution of $L(C_5H_8)$ through $NO_3$ oxidation (Figure 11B). Average $L(C_5H_8)$ through night-time $NO_3$ oxidation increased to $8.4 \times 10^{-5}$ ppbv s$^{-1}$, but $L(C_5H_8)$ through OH oxidation decreased to $9.2 \times 10^{-5}$ ppbv s$^{-1}$, resulting in a relatively high contribution of $NO_3$ oxidation (32 – 57%). $NO_3$ oxidation

contributed to 44% of the night-time $C_5H_8$ loss, which is comparable to OH oxidation (48%) and much higher than $O_3$ oxidation (8%). Considering that $C_5H_8$ is an important common hemiterpene emitted from multitudinous vegetations and its oxidation plays a key role in secondary organic aerosol (SOA) formation, daytime OH-induced $C_5H_8$ oxidation was highlighted while $NO_3$-induced oxidation of $C_5H_8$ may also significantly affect the SOA formation during the night-time (Brown and Stutz, 2012; Mellouki et al., 2021).





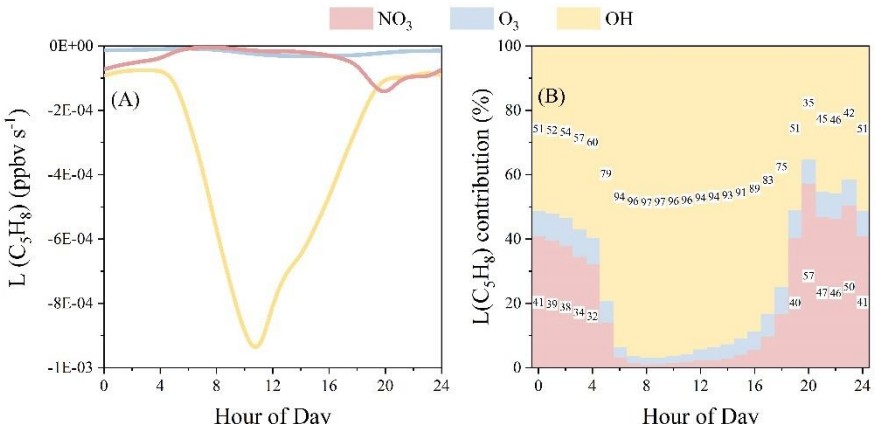


**Figure 11: (A): Loss rate of $C_5H_8$ through each path and (B): their relative contributions at each hour of the day.**

### 3.3.3.1 HNO₃ Formation

As an important component of particulate matter, inorganic nitrate (pNO₃) was produced through the partitioning of HNO₃. Hence, the production of HNO₃, defined as $P(HNO_3) = P(HNO_3)_{OH} + P(HNO_3)_{NO_3}$, represents the upper limits of pNO₃

production. $P(HNO_3)_{OH}$ denotes the HNO₃ production through (R-18) in the model (Sce-0). For $P(HNO_3)_{NO_3}$ calculation, both HNO₃ formation through N₂O₅ heterogeneous uptake on the aerosol surface (R-16) and other NO₃-induced reactions were considered (the former was the dominated one). In the model, parameterization for the heterogeneous N₂O₅ uptake is presented in (Eq-9).

$$NO_2 + OH \rightarrow HNO_3 \tag{R-18}$$

$$P(HNO_3)_{N_2O_5} = \frac{v(N_2O_5) \times S_a \times [N_2O_5]}{4} \times \gamma_{N_2O_5} \times 2, \tag{Eq-9}$$

where $v(N_2O_5)$, $[N_2O_5]$, and $\gamma_{N_2O_5}$ represent the molecular speed, concentration, and heterogeneous uptake coefficient of N₂O₅, respectively. $\gamma_{N_2O_5}$ was typically set as 0.1 reported in previous studies (Brown and Stutz, 2012; Wang et al., 2017).

As shown in Figure 12, the overall P(HNO₃) was high during the daytime and low during the night-time. During the daytime, $P(HNO_3)_{NO_3}$ was generally much lower than $P(HNO_3)_{OH}$, leading to high contributions of $P(HNO_3)_{OH}$ (>90%). However,

during the night-time, $P(HNO_3)_{OH}$ remarkably decreased but $P(HNO_3)_{NO_3}$ showed an increase, which promotes the relative contribution of $P(HNO_3)_{NO_3}$ to the sum P(HNO₃). On average throughout all day, $P(HNO_3)_{NO_3}$ contributed 18%, significantly lower than $P(HNO_3)_{OH}$ (82%). However, at night, $P(HNO_3)_{NO_3}$ contribution increased to 51%, slightly higher than $P(HNO_3)_{OH}$ (49%). By far, very few NO₃ measurements are available in China (Lu et al., 2019; Suhail et al., 2019), while its high concentration and important role in chemical oxidation presented in this study shed light on the necessity of direct NO₃ (as

well as related species such as N₂O₅, ClNO₂, etc.) measurements in the NCP, where summertime O₃ level is substantially increasing.

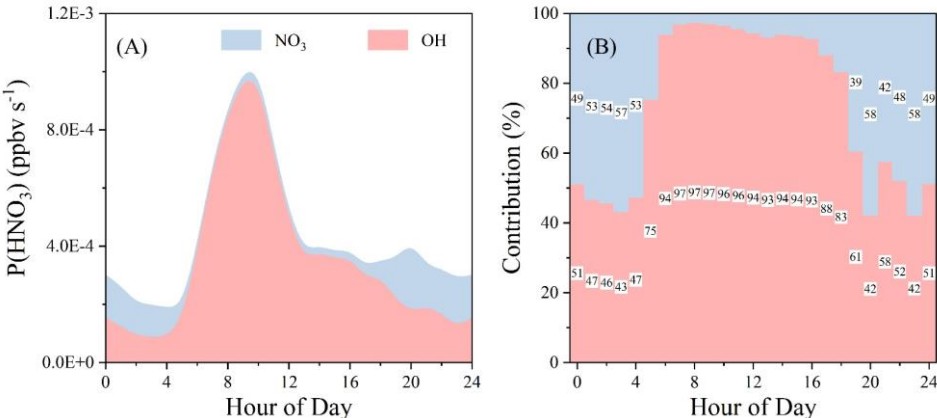

**Figure 12: (A): HNO₃ production (P(HNO₃)) from NO₃- or OH- induced reactions and (B): their relative contribution at each hour of the day. NO₃-induced reactions include heterogeneous uptake of N₂O₅ on the aerosol surface and all the other NO₃ reactions that produce HNO₃.**

### 3.3.4 Radical Initiation vs. Termination

Measurements on other radical precursors, such as $H_2O_2$ (through photolysis to produce OH), HCHO (through photolysis to produce $HO_2$), and alkenes (through ozonolysis via Criegee intermediate to produce OH and $HO_2$), were available, which allows a comparison of radical initiation (primary production) and termination ($T(RO_x)$). As shown in Figure 13, the overall radical initiation and termination showed similar variations and levels. Both, the sum of the $RO_x$ initiation and termination showed peaks of about $2.1 \times 10^{-3}$ ppbv s$^{-1}$ at noon, which are in the range of $0.7 - 3.4 \times 10^{-3}$ ppbv s$^{-1}$ reported in previous studies (Elshorbany et al., 2010, 2012; Hofzumahaus et al., 2009; Kukui et al., 2014; Liu et al., 2012; Ren et al., 2003). During the daytime, it is evident that HONO photolysis ($P(RO_x)_{HONO\_net}$) made the largest contribution (20 – 70%, Figure 13B) to $RO_x$ initiation, with an average of 37% (or 32% for all-day, Figure S11), followed by ozonolysis (29%), $O_3$ photolysis (21%), HCHO photolysis (13%), and $H_2O_2$ photolysis (1%). In particular, $RO_x$ production from the ozonolysis of alkenes was significantly lower than that from HONO during 6:00 – 14:00 until later after 17:00 when it started to dominate $RO_x$ production. At night with the absence of photochemistry, ozonolysis was the major source for primary $RO_x$ and exhibited similar levels to itself during the daytime, leading to its important role in primary $RO_x$ production (39% for all day). Besides, $T(RO_x)$ was dominated by $NO_2 + OH \rightarrow HNO_3$, $NO_2 + CH_3COO_2 \rightarrow PAN$, and $HO_2 + HO_2 \rightarrow H_2O_2$ (Elshorbany et al., 2010, 2012; Hofzumahaus et al., 2009; Kukui et al., 2014; Liu et al., 2012; Stone et al., 2012).



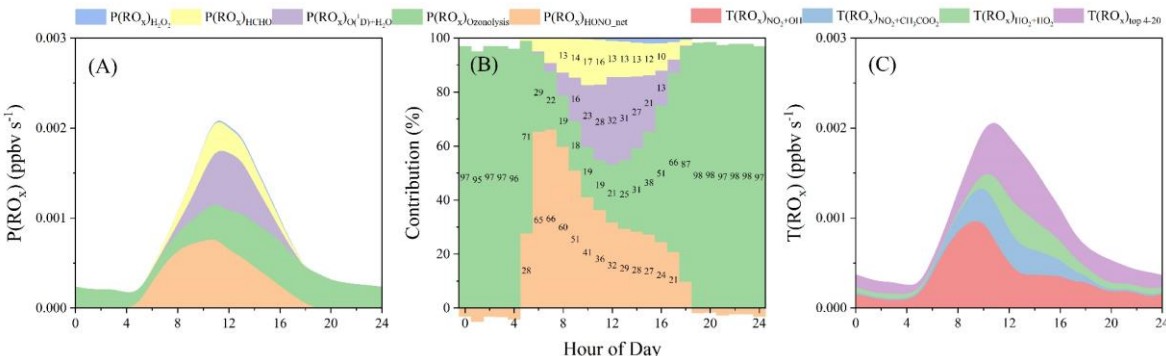

**Figure 13: (A): Primary RO$_x$ production from different sources and (B): their relative contributions at different hours of the day, and (C): the top-20 RO$_x$ loss rates. Note that: 1) due to an integration problem, the top-20 net radical loss paths were summarized here and it could represent the majority of total T(RO$_x$) as others (<1×10$^{-5}$ ppbv s$^{-1}$) were at least 2 orders of magnitude lower than the sum of top-20, 2) night-time P(RO$_x$)$_{HONO\_net}$ was negative (a net sink for OH) so that its contribution was also negative at night and 3) the same amounts of radical loss or production from equilibrium reactions (e.g., HO$_2$ + NO$_2$ ↔ HNO$_4$; CH$_3$COO$_2$ + NO$_2$ ↔ PAN) was excluded from radical initiation or termination.**

### 3.3.5 Role of HONO in OH Production at the Foot and the Summit Stations

Although measurements at the foot and the summit stations were conducted during two consecutive periods rather than the same one in summer 2018, it still allows a reasonable comparison of HONO contribution to OH formation at the foot station (lower boundary layer) and the summit station (upper boundary layer). Because of limited data available at the summit station, we only compared HONO with O$_3$ in primary OH formation. As reported in the companion paper, rapid vertical transport maintains the high HONO level at the summit station, promising HONO an important role in integrated OH production with a contribution of 26% to the sum of OH production (P(OH)$_{O_3+HONO}$) considering only HONO and O$_3$ photolysis. If OH loss through HONO + OH and NO + OH was subtracted from P(OH)$_{HONO}$ (then it becomes P(OH)$_{HONO\_net}$), its contribution decreased to 18%, about one-quarter of P(OH)$_{O_3}$.

Then net OH production from HONO (P(OH)$_{HONO\_net}$) and O$_3$ (P(OH)$_{O_3}$) photolysis at the foot and summit stations were also summarized and compared. As shown in Figure 14, it is apparent that HONO photolysis initialized the daytime photochemistry at both the foot and the summit stations as P(OH)$_{HONO\_net}$ dominated OH production in the early morning. Average P(OH)$_{HONO\_net}$ and P(OH)$_{O_3}$ at the foot station are 2.4×10$^{-4}$ and 1.4×10$^{-4}$ ppbv s$^{-1}$, respectively, both of which are significantly higher than those (1.7×10$^{-5}$ and 7.7×10$^{-5}$ ppbv s$^{-1}$) at the summit station as a result of relatively lower HONO and O$_3$ concentrations and lower solar photolysis frequencies observed at the summit station. The latter was caused by frequent cloud formation neat the summit. Nevertheless, the considerable contributions of P(OH)$_{HONO\_net}$ to P(OH)$_{O_3+HONO}$ at the foot (64%) and the summit (18%) stations indicate the essential role of HONO in the atmospheric oxidizing capacity at both the ground (lower boundary layer) and the summit (upper boundary layer) levels in mountainous regions.





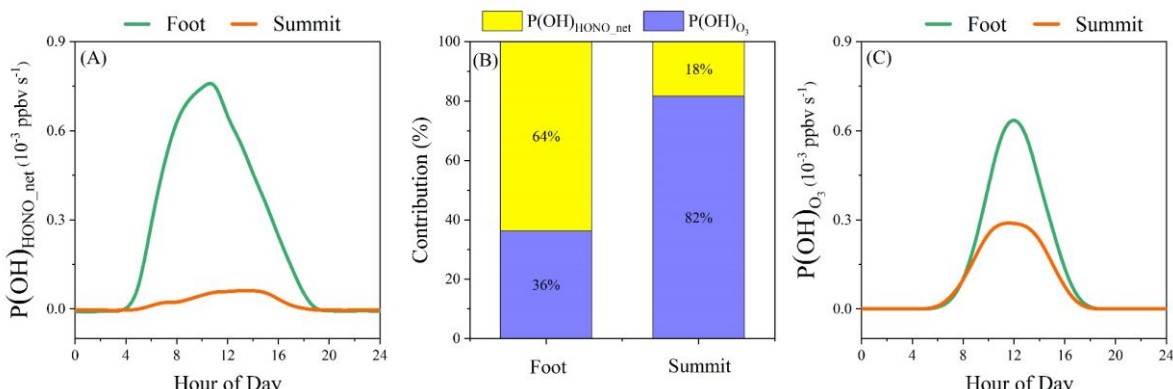

**Figure 14: (A): Net OH production from HONO (P(OH)$_{HONO\_net}$), (C): net OH production from O$_3$ (P(OH)$_{O_3}$) photolysis, and (B): their relative contributions at the foot and the summit of Mt. Tai.**

## 4. Summary

Atmospheric HONO and related parameters (VOCs, NO$_x$, PM$_{2.5}$, J(NO$_2$), etc.) were measured at the foot and the summit of Mt. Tai in the summer of 2018. The present study was conducted mainly based on measurements at the foot station. The observed HONO varied from 0.05 to about 3 ppbv, with an average of $0.62 \pm 0.42$ ppbv. With the implementation of a 0-D box model (F0AM) coupled with the Master Chemical Mechanism (MCM v3.3.1), the HONO budget and the radical (RO$_x$ + NO$_3$) chemistry were explored.

The main conclusions are summarized as follows:

1. The default HONO source, NO + OH, significantly underestimated the observed HONO by 87%, revealing a strong unknown source (P$_{unknown}$). The diurnal profile of P$_{unknown}$ rapidly increased in the morning and peaked nearly 3 ppbv h$^{-1}$ at noon, suggesting additional photo-enhanced HONO formation processes.

2. A HONO/NO$_x$ ratio of 0.7% was derived for direct emission, and its contribution (15 – 25% at night but negligible
during the daytime) was furtherly quantified by a new method developed in this study. Based on the constraints on the aerosol-derived HONO sources (NO$_2$ uptake on the aerosol surface and nitrate photolysis) obtained from the summit measurement (see the companion paper) and from recent laboratory studies, we found that the aerosol-derived HONO sources may make moderate or small contributions to HONO formation at the summit level and the ground level, respectively, but their contributions were not higher than NO + OH. Heterogeneous NO$_2$ conversion on the
ground surface made the largest contribution to P$_{unknown}$, but it was sensitive to the MLH used for its parameterization. This addressed the importance of a reasonable MLH for exploring ground-level HONO formation in 0-D models and the necessity of vertical measurements.

3. HONO played an important role in RO$_x$ but a negligible role in NO$_3$ concentrations. OH dominated the atmospheric oxidizing capacity in the daytime, while NO$_3$ appeared to be significant at night. Peaks of NO$_3$ time series and diurnal
variation reached 22 and 9 pptv, respectively. NO$_3$ induced reactions contribute 18% of nitrate formation potential

 

and 11% of the $C_5H_8$ oxidation throughout the whole day. While at night, $NO_3$ chemistry led to 51% or 44% of the nitrate formation potential or the $C_5H_8$ oxidation, respectively. $NO_3$ chemistry may significantly affect night-time secondary organic and inorganic aerosol formation in this high $O_3$ region. Hence, the direct measurement of $NO_3$ (along with $HO_x$, $N_2O_5$, $ClNO_2$, etc.) in this region should be conducted.

4.   A comparison of HONO contributions to primary OH at the summit and the ground levels was conducted and it was confirmed that HONO photolysis initialized daytime photo-chemistry at both sites in the early morning. On average, HONO made contributions of 64% and 18% to $P(OH)_{O_3+HONO}$ at the summit and the ground levels, respectively. As HONO observed at the summit level was mainly transported from the ground level, it addressed the role of HONO in the atmospheric oxidizing capacity in both the lower and the upper boundary layer over mountainous regions.

**Acknowledgment**

We are grateful to Shuyu Sun for her help with OVOCs measurements. We thank all researchers involved in this campaign from the Research Centre for Eco-Environmental Sciences-Chinese Academy of Sciences, Fudan University, Shandong Jianzhu University, Shandong University, and the Municipal Environmental Protection Bureau of Tai'an. C.X. thanks the University of Leeds for providing the MCM v3.3.1 and Glenn M. Wolfe for providing the F0AM platform.

**Funding**: This work was supported by the National Natural Science Foundation of China (Nos. 91544211, 41727805, 41931287, and 41975164), and the PIVOTS project provided by the Region Centre − Val de Loire (ARD 2020 program and CPER 2015 − 2020).

**Author Contribution**: C.X., C.Y., W.Z., X.H., P.L., C.Z., X.Z., C.L., Z.M., J.L., and J.W. performed the field measurements. C.X. analyzed the observation data, performed model simulations, and wrote the paper with input from all co-authors. C.Y. and J.K. also contributed by fruitful discussions and comments on model simulations and writing. C.Y., K.L., V.C., J.K., A.M., and Y.M. revised the manuscript.

**Competing Interests**: The authors declare no competing financial interest.

**Data Availability**: All the data used in this study is available upon request from the corresponding authors.

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
