# Peer review of "Atmospheric Measurements at Mt. Tai - Part II: HONO Budget and Radical (ROx + NO3) Chemistry in the Lower Boundary Layer"

_Atmospheric Chemistry and Physics, 2021_

## Author Comment (AC1)

**General comments:**

Tropospheric chemistry often begins with a trace gas molecule reacting with one of the tropospheric oxidants: OH radicals, $NO_3$ radicals, or $O_3$ molecules. (Reactions initiated by Cl atoms, generated from $ClNO_2$ photolysis, also contribute to the atmosphere's oxidising capacity, although Cl chemistry was not considered in this paper.) Such reactions determine the lifetime of traces gases in the atmosphere; this chemistry also leads to the production of secondary pollutants such as tropospheric ozone, organic nitrates (PAN etc) and secondary aerosol particles. Under most conditions, OH radicals make the largest contribution to the atmosphere's oxidising capacity. However there are gaps in our detailed understanding of OH sources, notably concerning the importance of the photolysis of nitrous acid (HONO).

This paper reports observations of HONO made in an urban location (Tai'an city) in the central North China plain over a period of several weeks in summer. The authors find that simplest "default" HONO source from the OH + NO reaction (which is not a net source of OH) only accounts for 13% of the observed HONO, averaged over the measurement campaign. Thus a large additional "unknown" HONO source must exist, and the authors explore 6 possibilities. The dominant HONO source at the measurement site is identified as the heterogeneous conversion of $NO_2$ into HONO on ground surfaces, either photo-enhanced by sunlight during the day (approx. 80% of the source strength), or without light in the dark at night (approx. 70%). Small contributions (of up to around 10% each) variously come from the conversion of $NO_2$ into HONO on aerosol during the day or at night, the photolysis of particulate nitrate, or direct emissions of HONO (e.g. from traffic or soils). The authors have done a good job to pick apart and characterize the strengths of these different possible HONO sources. This information is valuable to atmospheric chemists and modellers.

In the second half of the paper, the authors employ a zero dimensional box model to investigate what contribution OH derived from HONO makes to the radical chemistry at their measurement site. The model is used to model OH from HONO and from other sources (ozone photolysis, unsaturated VOCs reacting with ozone, etc), sources & sinks of $HO_2$ and $RO_2$ radicals, sources & sinks of $NO_3$, and the changing contributions of OH- and $NO_3$-initiated oxidation chemistry over the 24 hour cycle. The model's many outputs are nicely illustrated on well-constructed and informative plots. But this part of the work was less convincing due, primarily, to the simplicity of the 0-D model which cannot consider the vertical gradients in HONO concentrations that are known to exist from ground-based HONO sources. The authors are honest about the limitation of their modelling approach and say that a 1-D box model would provide better conclusions. The nighttime oxidant $NO_3$ and its reservoir $N_2O_5$ were not measured in this study, but modeled instead, and this leads to further uncertainties in assessing the relative contributions of OH and NO3 reactivity (as acknowledged by the authors).

Response: Thanks for your great efforts and valuable comments, which helps to improve our manuscript. Please see the point-to-point response below.

**(Comments in Black, Response in Blue, Changes in the manuscript in Red)**

Specific comments:

[Comment 1] The success (or otherwise) of the 0-D box model depends on making the right choice for HONO's mixing layer height (section 3.2.2.4, line 343). The authors choose 50 m for HONO's MLH because that generates the best agreement between modeled and measured HONO production rates, P(HONO), in Fig 6(b). But Figure 6(b) shows that setting MLH to 35 m instead of 50 m

increases P(HONO) by roughly a factor of 1.5; alternatively, setting MLH to 100m decreases P(HONO) by a factor of 2. Thus relatively small changes in the MLH generate big changes in modeling the ground HONO source (i.e. the dominant HONO source), with implications for understanding the OH source strength and radical chemistry. This raises questions that require answers:

* How much do this paper's conclusions depend on the precise choice of the MLH? – how would the conclusions change if the actual MLH were 35 m or 100 m or 200 m?

Response: We have to admit that MLH has a strong impact on modeling ground-derived HONO sources in a box model, for which a homogeneous MLH is assumed, i.e., with the same kinetics, an increasing MLH would cause a decreasing contribution of a ground-derived source. However, MLH does not affect a) aerosol-derived HONO sources (see the equations for related parametrizations in Sections 3.2.2.2 and 3.2.2.3), assuming no significant gradient of the particle concentration in the MLH or b) the main conclusions of this paper. Please see the detailed explanation in the response to the followed comment below.

* Is there a way to independently verify that the optimum choice of HONO's MLH is 50 m? – can this MLH value be proved by comparing HONO observations made at the foot site in Tai'an city with HONO measurements at the nearby summit of Mount Tai (using data from the companion paper)? Is the vertical gradient in HONO at this site consistent with other studies e.g. Brown et al (J GeoPhys Res, 118, 8067, 2013; doi:10.1002/jgrd.50537) who also observed vertical gradients in $NO_3$, $N_2O_5$ and $ClNO_2$?

Response: Due to the gradients of HONO observed in the atmosphere, near-ground surface HONO measurements are typically more weighted by ground-derived sources (Vandenboer et al., 2013; Vogel et al., 2003; Wong et al., 2011, 2013). As shown in a recent vertical HONO measurement in southwest China (Xing et al., 2021), HONO levels rapidly decreased from 4.8 ppbv at the ground level (~4 m above the ground surface) to 1.6, 0.7, 0.3, 0.2, and 0.1 ppbv averaged in height ranges of 0 – 100, 100 – 200, 200 – 300, 300 – 400, and 400 – 500 m above the ground level, respectively. This indicates that the near-ground HONO was remarkably affected by ground-derived sources and hence the MLH should be much less than 100 m in box model studies constrained by near-ground surface measurement data. A similar phenomenon could also be found in Brown et al. (2013) and Vandenboer et al. (2013) in which they used a tower-based platform to measure the vertical profile of HONO. Therefore, we chose 50 m as the MLH.

Comparison between HONO measurements at the foot and the summit could not help constrain MLH because HONO at the summit station was mainly originated from the vertical transport driven by mountain winds rather than typical atmospheric vertical turbulences (see the companion paper about the summit observations at https://acp.copernicus.org/preprints/acp-2021-529/#discussion).

To reduce the uncertainties caused by the assumed MLH, we also conducted a) several sensitivity tests with different MLH (35, 50, and 100 m, see Figure 6B and Section 3.2.2.4) and b) enlarged aerosol-derived sources to see whether the aerosol-derived sources could explain the observations if ground-derived sources were reduced. Results showed a significant discrepancy between the modeled HONO and the observations assuming aerosol-derived sources dominated HONO formation (Figures S8 and S9), which reinforced our main conclusion about the HONO budget.

Hence, the MLH used in this study may have uncertainties (honestly, this is still an open question for near-ground HONO studies), but does not change the main conclusions about the HONO budget.

Additionally, we also addressed the necessity of the vertical measurement based on tower-, balloon-, or aircraft-based platforms and the coupling with 1-D model simulations (See Section 3.2.2.4).

* How does HONO's MLH change with e.g. wind speed, time of day, daytime versus night? Was a constant MLH = 50 m assumed to model the HONO time series (Fig 7a) and HONO's average diurnal profile (Fig 7b)? How would variability in MLH affect this work's conclusions?

Response: HONO's MLH may vary with wind speed, time of day, daytime versus night, etc. however, without continuous vertical measurement, the variation of HONO's MLH could not be achieved. Please see the response to the previous comment for the use of MLH and related work we did to reduce its uncertainties.

A constant MLH (50 m for normal simulations and 35 or 100 m for sensitivity tests) was used for all the model simulations, including Figures 7a and 7b.

Please see the above response for the impact of MLH variability in the conclusions.

[Comment 2] Figures 1 and 2 show detailed time series of the observations, and Table 2 provides statistics (all good). The information that is missing is average diurnal profiles of NO, $NO_2$, $O_3$, J(HONO) etc – please add these plots either in section 3.1.1 or the supporting information. Please add the diurnal profile of J(HONO) to one or all of Fig 4, Fig 6 and Fig 7(b) [and Fig S7(b) and Fig S8(b)], so that reader can easily distinguish day vs night, sunrise and sunset, photolytic vs dark HONO sources.

Another reason why I request the diurnal plots of NO, $NO_2$, $O_3$ and J(HONO) etc is because the diurnal plot of P(HONO) is not symmetric around 12:00 noon (Fig 4 and Fig 6). The unknown HONO source is faster in the morning than the afternoon, and peaks at 11am. The authors note this asymmetry on line 259 but they don't explain it. What is the cause of this asymmetry in HONO? And is P(HONO) expected to also be asymmetric at other locations? [I'm guessing it's not due asymmetry in J(HONO) because P(OH) from ozone photolysis *is* symmetric around 12:00 noon in Fig 14c.]

Response: We added diurnal variations of HONO, $O_3$, $NO_2$, J($NO_2$), $PM_{2.5}$, CO, NO, and $SO_2$ in the supporting information (Figure S3) and J(HONO) in Figure 4, which are shown below:

[Figure]

Figure S3: Diurnal profiles of HONO and related species measured during this campaign. Note that

some of these data were also shown in the companion ACP paper for comparison between measurements at the foot and the summit stations.

[Figure]

Figure 4: Simulated HONO by the default mechanism (Sce-1, left axis) compared with the observations (Obs, left axis), unknown source strength ($P_{unknown}$, right axis), HONO photolysis frequency (J(HONO), right axis), and the relative contributions of NO + OH to the observations at night (19:00 – 4:50, left pie chart) and day (5:00 – 18:50, right pie chart), respectively. The shaded area in blue represents the difference between the observation and modeled values.

Regarding the asymmetric variation of HONO and $P_{unknown}$, in Section 3.2.1, we improved the the related text as "Note that the profile of $P_{unknown}$ was asymmetric around 11:00, indicating the unknown source is not simply photolytic but also includes its precursors that also have an asymmetric distribution (e.g., $NO_2$, Figure S3)." Besides, we found that the scenarios with enlarged aerosol-derived sources showed worse performance in reproducing this asymmetry, which could be another piece of evidence that aerosol-derived sources are not the dominant ones. We then improved related text in Section 3.2.3 as "While that the modified model could also generally predict the observed HONO levels (Figures S8A and S9A) but it largely failed to reproduce the noontime observations in levels and variations (Figures S8B and S9B) including its asymmetry as mentioned in Section 3.2.1. This observation reinforces our conclusion that aerosol-derived sources play only a minor role in the daytime HONO formation."

[Comment 3] How do the conclusions of this paper translate to other locations in China and other countries? In terms of air quality, this is a polluted site, so are conditions here somehow special? For example, because high [$NO_2$] produces an especially large ground-based HONO source at this site? And/or because high ozone, [$O_3$] > 100 ppbv, is reacting rapidly with unsaturated VOCs to produce an especially active $HO_2$ and $RO_2$ chemistry at this site? For example, $NO_3$ is traditionally seen as the dominant nighttime oxidant, yet the modeling here in this study shows the sink for isoprene ($C_5H_8$) at night is approximately 50% to OH and 50% to $NO_3$ – it is unusual to see such active OH reactions at night. [BTW: I agree with the authors that it is better to measure $NO_3$ than to model it.]

Response: Generally, we believe that the parametrizations for HONO sources could be broadly used for near-ground measurements worldwide and similar conclusions are also expected in polluted

regions. For instance, our previous study (Xue et al., 2020) found similar conclusions in a pilluted site in the winter North China Plain, where air pollution (aerosol, $NO_2$, HONO, etc.) is much higher than that at this site. It is noteworthy that the method for quantifying direct HONO emission, which is firstly proposed in this study, could significantly reduce the overestimation of direct HONO emission during daytime. We suggest further studies to use this method to have more accurate results. At this site, high $O_3$ levels were observed both during the daytime and the night-time (Figure S3D), which is typical for the increasing $O_3$ pollution in the North China Plain (Han et al., 2020; Li et al., 2019; Sun et al., 2016, 2019). Hence reactions of $O_3$ with unsaturated VOCs play important role in OH production, particularly during the night-time maintaining and dominating OH production (Figures 9A and 13A).

As OH and $NO_3$ measurements are not available, we can only use the model results to shed light on the atmospheric chemistry in this polluted region and point out the uncertainties. A related text in Section 3.3.3.2 was improved as "Model results may have uncertainties but shed light on the atmospheric chemistry in this polluted region. By far, very few $NO_3$ measurements are available in China (Lu et al., 2019; Suhail et al., 2019), while its high concentration and important role in chemical oxidation presented in this study indicate the necessity of direct $NO_3$ (as well as related species such as $N_2O_5$, $ClNO_2$, etc.) measurements in the NCP, where summertime $O_3$ levels are substantially increasing (Han et al., 2020; Li et al., 2019; Sun et al., 2016, 2019)."

**Technical corrections:**

Generally, the paper is written well. However, some of the sentence structure is long and complex, especially those sentences that list multiple measurements, ideas or findings. Consider rewording and/or breaking into separate sentences [e.g. lines 40-43; line 132 (see below)].

Response: We separated Lines 40-43 as two sentences: "HONO photolysis in the early morning initialized the daytime photochemistry at the foot station. It was also a substantial radical source throughout the daytime, with contributions higher than $O_3$ photolysis to OH initiation."

Line 132 was improved as "10-min average data (except for $PM_{2.5}$) were used for the analysis of time series and statistic descriptions of the data. In contrast, hourly data were used for model simulations."

In addition, the English of the manuscript was carefully improved by the authors with the help of comments from three reviewers (See the response to comments and the manuscript).

Some figure captions also have long and complex sentence structure; they likewise could be improved by breaking them into several shorter sentences, e.g. Fig 9 "Simulated concentrations of (A) OH… and (D) NO3. [New sentence] The different coloured lines show…".

Response: Captions for Figure 9 were modified as "Simulated concentrations of (A): OH, (B): HO2, (C): RO2, and (D): NO3. Different colored lines show results in different scenarios (NO + OH: only with the homogeneous source; Base: constrained by the observed HONO; +50%: constrained by the observed HONO×1.5; +100%: constrained by the observed HONO×2; -50%: constrained by the observed HONO×0.5; -100%: constrained by HONO = 0)."

Besides, other figure captions were also improved similarly as the reviewer suggested.

Fig 13: the 3 notes in the figure caption would work better as 3 separate sentences.

Response: Improved as suggested.

Fig 4: caption needs separate sentences to explain the main panel and the pie charts.
Response: Improved as suggested.

 Abstract line 32: "Our model's default HONO source from the OH + NO reaction could only reproduce 13% of the observed HONO…"
Response: Improved as suggested.

Abstract line 37: "A HONO/NOx ratio of 0.7% from direct emissions [plural] was inferred…". Also please state what are the direct sources of HONO at this measurement site?
Response: This line was improved as "A $\Delta HONO/\Delta NO_x$ ratio of 0.7% for direct emissions from vehicle exhaust was inferred and a new method to quantify its contribution to the observations was proposed and discussed."

Abstract line 43: State clearly that $NO_3$ concentrations come from the 0-D model. Otherwise, without reading the main body of the paper, readers might assume $NO_3$ was measured.
Response: We added "modeled" before $NO_3$.

Abstract lines 46-48: the sentence repeats itself.
Response: the repeated part was deleted. "At night, $NO_3$ chemistry led to 51% or 44% of P($HNO_3$) or the $C_5H_8$ oxidation, respectively.  implying that $NO_3$ chemistry could significantly affect night-time secondary organic and inorganic aerosol formation in this high-$O_3$ region."

Line 79: NO3 radicals [plural]
Response: Improved as "the $NO_3$ radical".

Line 99: However, very few observations have been reported of OH and NO3 concentrations…
Response: Improved as suggested.

Line 115: Mt Tai is located in…
Response: Improved as suggested.

Line 117 / end of section 2.1.1: Add a sentence to explain the scientific motivation for making HONO observations at this ground site and at the summit of Mt Tai, and briefly explain what the authors found. Readers should be able to know this without having to find and read the companion paper.
Response: Thanks for the suggestion. We added a sentence: "Measurements at these two stations allow us to study HONO formation and its role in the atmospheric oxidizing capacity of the lower (the foot study) and the upper boundary layer (the summit study). Briefly, in the summit study, we found remarkably high daytime HONO levels as well as high unknown HONO source strength, which was mainly caused by rapid vertical transport from the ground to the summit levels driven by mountain winds."

Line 120-121: Here it states the LOPAP measurements took place in 2017. But line 109 and the time axis labels in Fig 1 and 2 show 2018. Which is correct?

Response: Sorry about this mistake. The campaign was taken place in the summer of 2018. We corrected the text in the manuscript.

Line 132: Reword: 10 minute data were used in X. [New sentence] Hourly data were used in Y.

Response: Improved as suggested.

Line 133: PM2.5 measurements were obtained… [plural]

Response: Improved as suggested.

Line 136: SZA functions [plural]

Response: Improved as suggested.

Line 144: interferences in [not "of"] the NO2 measurements [plural]

Response: Improved as suggested.

Line 169-171: Air masses [plural] observed at this site [delete "was"] originated from… which are shown in the wind rose plot in Figure S2.

Response: Improved as suggested.

Line 256: HONO mixing ratio differences… were calculated… and they are compared with P_unknown in Figure S5. [plurals]

Response: Improved as suggested.

Line 257: What do the authors mean by the word "typical"? – the average diurnal profile of P_unknown inferred from their HONO observations? I doubt P_unknown shown in Fig 4 can typically be applied to all other measurement sites.

Response: The sentence was deleted.

Line 278: Add a reference to the recent study of HONO emissions from traffic in a road tunnel by Kramer et al, who found a similar HONO/NOx ratio of 0.85%. https://doi.org/10.5194/acp-20-5231-2020.

Response: Thanks for the information. We then added this reference.

Line 291 / Table 4: Why are there two values listed for HONO (and two values for NO and two values for NOx)? Are they the HONO concentrations at the start and end of each plume of fresh emissions? Also units = ppbv?

Response: We added a sentence in the caption of Table 4: "Concentrations (ppbv) of HONO/NO/NO$_x$ at the start and the end of each plume of fresh emissions are also shown."

Line 302: What is meant by "overestimated one" and "popularly used one"?

Response: The "overestimated one" refers to the $\gamma_a$ value inferred from the summit study. Briefly,

we assumed that all the unknown HONO sources originated from $NO_2$ uptake on the aerosol surface then we could get a $\gamma_a$ value that constrained its upper limit (details can be found in the companion paper). The "popularly used one" refers to the $\gamma_a$ value obtained from laboratory flow tube experiments (Stemmler et al., 2006, 2007), which is popularly used in model studies.
We added related references in the manuscript.

Line 340: "reflected by" is not the right verb. [I'm not sure what the correct verb should be because I didn't understand this sentence.]
Response: "reflected by" was replaced by "described by".

Line 355: Therefore, the MLH for HONO was set at 50 m, with sensitivity tests performed with the MLH set at 35 and 100 m.
Response: This sentence was improved as "Therefore, the MLH for HONO was set at a constant height of 50 m, with sensitivity tests performed with the MLH set at 35 and 100 m."

Line 371: the model with the present HONO source parameterizations performed well in predicting… [the original adjective "magnificent" is too strong].
Response: Improved as suggested.

Line 378: reinforcing our conclusion that aerosol-derived sources played only a minor role in daytime HONO formation.
Response: Improved as suggested.

Line 400: rather than "The former one..,", it is clearer simply to begin the sentence with "HONO emissions from soils may occur…"
Response: Improved as suggested.

Line 447 / Fig 10: I found it difficult (& sometime impossible) to distinguish the different OH sources and sinks, where two shades of the same colour are plotted next to each other. The rainbow of colours is visually pretty, but the plot is impossible to interpret. Likewise P(NO3) and L(NO3).
Response: Figure 10 and its caption were improved as follows:

[Figure]

Figure 1: Production rates (P), loss rates (L) and reactivities of radicals. (A): L and P of OH; (B): L and P of NO$_3$; (C): Reactivities of OH and (D): Reactivities of NO$_3$. In (A) and (B), the top-3 sources or sinks are shown, and all the others are summarized in "Other sources" or "Other sinks". In (C), OH reactivities with different families of the measured species are shown and reactivities with all the unmeasured species are summarized in "Others". In (D), NO$_3$ reactivities from top-3 reactions are shown and all the others are summarized in "Others".

Line 447 / Fig 10: Why is NO$_3$ photolysis (during the day) not plotted in the NO$_3$ reactivity plot, panel (D)? Or is photolysis included in the "inorganic" reactions (orange colour)?

Response: Figure 10 was improved (see above) and NO$_3$ photolysis was shown in the graph.

Line 476: hemiterpene emitted by very many species of vegetation…

Response: Improved as suggested.

Line 483 / section 3.3.3.1: In addition to forming particulate nitrate, it should also be noted that HNO3 formation from OH+NO2 and from N2O5 hydrolysis on aqueous aerosol are also the major daytime and nighttime sinks for removing NOx from the atmosphere. Without these NOx sinks, NOx photochemistry would produce tropospheric ozone even more rapidly.

Response: We added this sentence: "HNO$_3$ formation from OH+NO$_2$ and from N$_2$O$_5$ hydrolysis on the aqueous aerosol surface are also the major daytime and night-time sinks for removing NO$_x$ from the atmosphere."

Line 487: the dominant one

Response: Improved as suggested.

Line 556: The default HONO source, NO + OH, significantly underestimated the observed HONO concentrations. [New sentence] This reaction could only account for 13% of the observed HONO, revealing a strong…"

Response: Improved as suggested.

**References**

[revised manuscript text omitted]

---

## Author Comment (AC2)

General comments:

HONO and related parameters were measured at the foot of Mt. Tai in the summer 2018. 0-D box model coupled with the MCM were used to explore the budget of HONO, OH, ROx and NO3 radical chemistry. The homogeneous reaction of NO and OH has been adopted as the default HONO source in the box model and account for 12%-15%. The family constraint was used in this Model scenario to correct for interferences of NO2 measurements. Large amount of unknown source of HONO appeared especially on the noontime. Then many sources of HONO were discussed and added in the models. Corrected NO2, direct emission, heterogeneous reactions of NO2, and photolysis reactions were considered in the model. Another part of the manuscript studied the Radical chemistry. The authors gave very detailed consideration on the sources of HONO, and some corrected methods were suggested. These results is meaningful for the development of HONO investigation.

Response: Thanks for your great efforts and valuable comments, which helps to improve our manuscript. Please see the point-to-point response below.

**(Comments in Black, Response in Blue, Changes in the manuscript in Red)**

There also existed some problems the authors need to improve the manuscript. The manuscript had two parts, one was about the sources of HONO, the other was about the radical chemistry. The connection between these two parts was not very tightly. The first part, more focused on the sources of HONO which had some relationship with OH radical, but how about NO3? I suggested the authors gave some descriptions on the connection between these two parts. For example, the significant of first part was that model was corrected more accuracy and could give more accurate results of radicals, such as ROx, NO3? Some relationship of HONO in NO3 chemistry?

Response: We added two sentences in Section 3.3 to strengthen the connection between these two parts: "Comprehensive field measurements in comparison to model studies allow studying the role of HONO in the radical chemistry of the atmosphere. HONO is expected to strongly impact OH levels in the lower atmosphere due to strong daytime HONO sources and due to its fast photolysis. In addition, considering high $O_3$ levels at the present field site, $NO_3$ chemistry could also be important particularly during night-time, which will also be discussed in this section."

Specific comments:

The logical of Introduction was not very well. The authors should give more discussions between the relationship of the investigation of HONO sources and radical (ROx + NO3) chemistry.

Response: See the above explanation about the connection between these two parts.

More detailed information of foot site should be presented especially the real environment around the site, which were very useful for the analysis of HONO sources.

Response: Locations of the foot and the summit sites could be found in Figure 1 in the companion paper. Besides, we add additional information about this site: "Tai'an is located nearly in the middle between Beijing and Shanghai. The city has a population of about 5.6 million and is about 60 km south of Jinan city (the capital city of Shandong province, population: ~8.7 million). Measurements were conducted both at the ground level (the foot of Mt. Tai, 150 m a.s.l.) and the summit level (the summit of Mt. Tai, 1534 m a.s.l., 36.23°N, 117.11°E). The foot station was inside Shandong College of Electric Power (36.18°N, 117.11°E) in the Taishan district of Tai'an city. There are no industrial

activities around this site, which is surrounded by the campus, residential area, and a business district. The 801$^{st}$ province road is in the northeast of this typical urban site."

In 3.1.2. Since the NO2 concentration is not credible by using thermo 42i, how did authors prove that the model results of NO2 correction were reliable. Additionally, the interference could be as high as +75% after adding HNO3 in model simulation, which corrected NO2 was used, consider HNO3 or not? If not, please give the explanation on why not considered HNO3 interference? By the model results, PAN had most impact on the NO2 concentration, how accuracy about the model results of PAN, have compared with observation PAN?

Response: HNO$_3$ was not considered. HNO$_3$ is highly sticky so that it is expected to be absorbed on the wall of sampling tubes (about 2 m) or on the filter before reaching inside of the instrument. Nevertheless, we included HNO$_3$ in an additional scenario for uncertainty analysis.

Regarding model results of PAN, we only have 10 days of PAN measurement in this model period. During these 10 days, diurnal profiles of the measured NO$_2$, modeled PAN, and the measured PAN were shown in the following figure. The model could well reproduce PAN in the daytime but there is underestimation during the night-time. Therefore, using the modeled PAN to correct NO$_2$ could improve the accuracy of daytime NO$_2$. Since night-time NO$_2$ was more than one order of magnitude higher than PAN, the underestimation of PAN during the night-time would not cause significant error in NO$_2$.

In summary, we have to admit that this method still has uncertainties but indeed reduced the overestimation in NO$_2$ measured by the chemiluminescence technique.

[Figure]

Figure: Diurnal profiles of the measured NO$_2$, modeled PAN, and the measured PAN.

What's the meaning of Fig S3? NOz*??? Line 223, also NOz*?

Response: There is a spelling mistake. NO$_z$* should be NO$_2$* defined as NO$_2$* = NO$_2$ + HNO$_4$ + 2N$_2$O$_5$ + NO$_3$ + PANs + organic nitrates* (see Section 3.1.2). We did the correction in the manuscript and the supporting information.

In 3.2.2.1. What was the correlation between ΔHONO/ΔNOx data in table 4 and HONO/NOx data in Fig 5. In Fig 5, the phenomenon of "the observed HONO/NOx is convergentas NO/NO2 increases" was unclear, this was not convincing for the further correction on ΔHONO/Δ NOx. There were definitely different meanings for ΔHONO/ΔNOx and HONO/NOx, why authors choose NOx

concentrations?

Response: "HONO/NO$_x$" appeared in Line 269 in this section was a spelling mistake. It should also be "ΔHONO/ΔNO$_x$" which represents the emission ratio from vehicle exhaust.

We chose NO$_x$ (NO+NO$_2$) as many previous studies did because 1) a rapid NO increase indicates a fresh plume emitted by vehicles and 2) NO-to-NO$_2$ conversion is fast but NO$_x$ doesn't change.

Give the explanation of why HONO from direct emission (HONOemi) is likely significantly overestimated with a constant ΔHONO/ΔNOx because of different lifetimes of HONO (τ(HONO)) and NOx (τ(NOx)) in the daytime. Please give the more reasonable explanation of the modified factor of $\frac{\tau(HONO)}{\tau(NO_x)}$ in equation 3, and detail information on the calculation of $\frac{\tau(HONO)}{\tau(NO_x)}$.

Response: In the daytime, NO$_x$ and HONO have distinctly different lifetimes as shown in Figure S7. For instance, at noon, the lifetime of NO$_x$ is about 4.4 h, which means the measured NO$_x$ at noon represents an accumulation of NO$_x$ emission during >4.4 h. However, the lifetime of HONO at noon is about 0.2 h, which means HONO has much less accumulation. Therefore, if we want to calculate HONO emission from vehicle exhaust, the accumulation time (i.e., lifetime) of HONO and NO$_x$ should be taken into consideration.

τ(HONO), could be obtained from HONO concentration divided by its total loss rates (HONO+hv and HONO+OH), which could be directly achieved from the F0AM model (Wolfe et al., 2016).

τ(NO$_x$) depends on the lifetime of NO$_2$ (τ(NO$_2$)) and NO/NO$_2$ ratio regarding the net loss of NO$_x$ is mainly in the form of HNO$_3$ produced through OH or NO$_3$ induced reactions. Similar to τ(HONO), we can calculate τ(NO$_2$) through NO$_2$ concentration divided by its net loss rate (NO$_2$ + OH → HNO$_3$, NO$_3$ + VOCs → HNO$_3$, and NO$_3$ + NO$_2$ + wet surface → HNO$_3$). Then we can get τ(NO$_x$) through the following equation (Seinfeld and Pandis, 2016):

$$\tau(NO_x) = \tau(NO_2) * (1 + \frac{NO}{NO_2})$$

Related texts were presented in Section 1 in the supporting information.

The observation site is special, how to choose NO2 uptake coefficient on aerosol surfaces and ground surfaces? what's the reasonable? Why the γa was larger than γa_dark? As shown in Eq-5, photo-enhanced effects had been considered. Similar question also appeared on the γg and γg_dark. Please give the explanation for the higher value of γa and γg.

Response: γ$_{a\_dark}$ and γ$_{g\_dark}$ represent NO$_2$ uptake coefficients on the aerosol surface and ground surface, respectively. γ$_{a\_dark}$ and γ$_{g\_dark}$ were found to be low, generally less than few $10^{-6}$ (George et al., 2005; Han et al., 2016; Kurtenbach et al., 2001; Ndour et al., 2008; Stemmler et al., 2006, 2007). NO$_2$ uptake could be enhanced by radiation, so photo-enhanced NO$_2$ uptake coefficients on the aerosol surface and ground surface (γ$_a$ and γ$_g$) are much higher than γ$_{a\_dark}$ and γ$_{g\_dark}$, which could be obtained from laboratory studies and widely used in field/model studies (George et al., 2005; Han et al., 2016; Stemmler et al., 2006, 2007).

MLH values have great impact on the simulation results, so the reasonable MLH value was very important. Why 50 m was good? please combined the real environment and give the reasonable discussions.

Response: Near-ground HONO measurements are typically more weighted by ground-derived

sources (Vandenboer et al., 2013; Vogel et al., 2003; Wong et al., 2011, 2013). As shown in a recent vertical HONO measurement in southwest China (Xing et al., 2021), HONO levels rapidly decreased from 4.8 ppbv at the ground level (~4 m above the ground surface) to 1.6, 0.7, 0.3, 0.2, and 0.1 ppbv averaged in height ranges of 0 – 100, 100 – 200, 200 – 300, 300 – 400, and 400 – 500 m above the ground level, respectively. This indicates that the near-ground HONO was remarkably affected by ground-derived sources and hence MLH should be much less than 100 m in box model studies constrained by near-ground surface measurement data. A similar phenomenon could also be found in Brown et al. (2013) and Vandenboer et al. (2013) in which they used a tower-based platform to measure the vertical profile of HONO. Therefore, we chose 50 m as the MLH.

To reduce the uncertainties caused by the assumed MLH, we also conducted a) several sensitivity tests with different MLH (35, 50, and 100 m, see Figure 6B and Section 3.2.2.4) and b) enlarged aerosol-derived sources to see whether the aerosol-derived sources could explain the observations if ground-derived sources were reduced. Results showed a significant discrepancy between the modeled HONO and the observations assuming aerosol-derived sources dominated HONO formation (Figures S8 and S9), which reinforced our main conclusion about the HONO budget.

Hence, the MLH used in this study may have uncertainties (honestly, this is still an open question for near-ground HONO studies), but does not change the main conclusions about the HONO budget. Additionally, we also addressed the necessity of the vertical measurement based on tower-, balloon-, or aircraft-based platforms and the coupling with 1-D model simulations (See Section 3.2.2.4).

Line 380: how HONOemi was included in the model? The HONOemi was not the production rate data by Eq-2 and Eq-3.

Response: $HONO_{emi}$ was calculated based on Eq-2 for nighttime and Eq-3 for night-time with consideration of different lifetimes of HONO and $NO_x$ during daytime. Details can be found in the above response.

In Figure 7, to include direct emission in the mode result, we added $HONO_{emi}$ (concentration in ppbv) to the modeled HONO (concentration in ppbv) to compare with the observations and then calculate its contribution.

To make it clear, we improved the captions in the figure as follows:

[Figure]

**Figure 7: Modeled HONO mixing ratios (Model, in blue) in comparison with observations (Obs, in black). (A): time series; (B): diurnal variations. Model+HONO$_{emi}$ represents the sum of the modeled HONO and HONO$_{emi}$.**

Line 534: what's the percentage of HONO contribution to OH radical not considering only HONO and O3 photolysis? From Fig 10, there were many sources in production of OH, and HONO not the most important sources.

Response: In Figure 10, all the OH sources (initiation and propagation) are included. Some of them are not primary OH sources but radical propagation sources (such as HO$_2$→OH, etc.). However, atmospheric chemistry is initialized by primary OH from initiation sources. Hence, we need to compare the relative contributions of primary OH sources, which leads to Section 3.3.4 and Figure 13.

I can't understand why put the foot and summit of Mt. Tai together in the title, through the two part manuscripts, the Part I was the results on the summit of Mt. Tai, while Part II was the results on the foot of Mt. Tai. the comparation of these two sites was only given some discussions in this manuscript 3.3.5, but these discussions had no new sights and meaning. Furthermore, the analysis methods were different in these two parts. I suggested the authors revised the title, this manuscript was "Atmospheric Measurements at Mt. Tai-Part II: HONO Budget and Radical (ROx + NO3) Chemistry in the Lower Boundary Layer".

Response: After careful consideration of the comments on the two papers, we decided to move Section 3.3.5 (Role of HONO in OH Production at the Foot and the Summit Stations) to the summit paper because the most important of this section is to highlight the important role of HONO in the summit level.

Besides, the titles of these two papers are modified as suggested.

**References**

George, C., Strekowski, R. S., Kleffmann, J., Stemmler, K. and Ammann, M.: Photoenhanced uptake of gaseous NO$_2$ on solid organic compounds: a photochemical source of HONO?, Faraday Discuss., 130, 195–210, doi:10.1039/b417888m, 2005.

Han, C., Yang, W., Wu, Q., Yang, H. and Xue, X.: Heterogeneous Photochemical Conversion of NO$_2$ to HONO on the Humic Acid Surface under Simulated Sunlight, Environ. Sci. Technol., 50(10), 5017–5023, doi:10.1021/acs.est.5b05101, 2016.

Kurtenbach, R., Becker, K. H., Gomes, J. A. G., Kleffmann, J., Lörzer, J. C., Spittler, M., Wiesen, P., Ackermann, R., Geyer, A. and Platt, U.: Investigations of emissions and heterogeneous formation of HONO in a road traffic tunnel, Atmos. Environ., 35(20), 3385–3394, doi:10.1016/S1352-2310(01)00138-8, 2001.

Ndour, M., D'Anna, B., George, C., Ka, O., Balkanski, Y., Kleffmann, J., Stemmler, K. and Ammann, M.: Photoenhanced uptake of NO2 on mineral dust: Laboratory experiments and model simulations, Geophys. Res. Lett., 35(5), L05812, doi:10.1029/2007GL032006, 2008.

Seinfeld, J. H. and Pandis, S. N.: Atmospheric Chemistry and Physics: From Air Pollution to Climate Change, John Wiley & Sons., 2016.

Stemmler, K., Ammann, M., Donders, C., Kleffmann, J. and George, C.: Photosensitized reduction of nitrogen dioxide on humic acid as a source of nitrous acid, Nature, 440(7081), 195–198,

doi:10.1038/nature04603, 2006.

Stemmler, K., Ndour, M., Elshorbany, Y., Kleffmann, J., D'Anna, B., George, C., Bohn, B. and Ammann, M.: Light induced conversion of nitrogen dioxide into nitrous acid on submicron humic acid aerosol, Atmos. Chem. Phys., 7(16), 4237–4248, doi:10.5194/acp-7-4237-2007, 2007.

Wolfe, G. M., Marvin, M. R., Roberts, S. J., Travis, K. R. and Liao, J.: The framework for 0-D atmospheric modeling (F0AM) v3.1, Geosci. Model Dev., 9(9), 3309–3319, doi:10.5194/gmd-9-3309-2016, 2016.

---

## Author Comment (AC3)

General comments:

Firstly, the English used in the manuscript is, at times, quite poor and it is recommended that the authors make use of an English-language proofreading service to tidy up the manuscript. However, having said that, the scientific content of the manuscript is generally fine and, after following the specific recommendations below, I would fully support its publication in ACP. One other general comment relates to the consistency of units: please standardise reaction rates with the widely-used ppbv h-1 (sometimes ppbv s-1 is used instead).

Response: Thanks for the comments, which help to improve our manuscript. Please see the point-to-point response below. **(Comments in Black, Response in Blue, Changes in the manuscript in Red)**

We carefully read this manuscript and have improved its English. Besides, comments from the three reviewers (particularly Reviewer 1) also helped a lot to improve the English of this manuscript.

Regarding the units, we unified them as suggested. Modified figures include: Figures 10, 11, 12, and 13.

Specific comments:

Figure 2 – it would be good to see diurnal profiles of the same key species, especially considering almost all of the other results are presented as diurnal profiles rather than full time series.

Response: We added diurnal variations of HONO, $O_3$, $NO_2$, $JNO_2$, $PM_{2.5}$, CO, NO, and $SO_2$ in the supporting information (Figure S3), which are shown below:

[Figure]

Figure S3: Diurnal profiles of HONO and related species measured during this campaign. Note that some of these data were also shown in the companion ACP paper for comparison between measurements at the foot and the summit stations.

Figure 10 – in A and B, the gradient-style colouring makes it difficult to distinguish the categories, please use more contrasting colours as in C and D. Also, for C and D, the caption states that unmeasured species are summarised in the "other" category. Does this refer to intermediates (e.g., OVOCs) generated in the model? i.e., model-generated species that were not measured but do contribute to the model reactivity? Please clarify.

Response: To clarify this issue, Figure 10 and its caption were improved as follows:

[Figure]

Figure 1: Production rates (P), loss rates (L) and reactivities of radicals. (A): L and P of OH; (B): L and P of NO₃; (C): Reactivities of OH and (D): Reactivities of NO₃. In (A) and (B), the top-3 sources or sinks are shown, and all the others are summarized in "Other sources" or "Other sinks". In (C), OH reactivities with different families of the measured species are shown and reactivities with all the unmeasured species are summarized in "Others". In (D), NO₃ reactivities from top-3 reactions are shown and all the others are summarized in "Others".

Figure 13 – in the caption, what is meant by the "integration problem"? I cannot find it mentioned in the main text. In general, the caption could do with rewording as it is not the most clear. Also, the caption states that equilibrium reactions were not considered, yet (net) PAN formation is shown in C?

Response: We didn't figure out how to integrate all the reactions that consume radicals. Therefore we summarized the top-20 radical termination reactions. The sum of the top-20 could represent the majority of total $T(RO_x)$ as others (<0.03 ppbv h$^{-1}$) were at least 2 orders of magnitude lower.

Net PAN formation is a net radical loss path and was shown in Figure 13C. It could be derived from PAN production rate through reaction (1) subtracted by its loss rates through reaction (2).

$$CH3COO2 + NO2 = PAN ; \quad (1)$$

$$PAN = CH3COO2 + NO2 ; \quad (2)$$

The updated figure and caption are as follows:

[Figure]

Figure 13: Primary $RO_x$ production and net $RO_x$ loss. (A): Production from different sources and (B): their relative contributions at different hours of the day. (C): the top-20 $RO_x$ loss rates. Note that the top-20 net radical loss paths were summarized here. It could represent the majority of total $T(RO_x)$ as others (<0.03 ppbv h$^{-1}$) were at least 2 orders of magnitude lower than the sum of top-20. Night-time $P(RO_x)_{HONO\_net}$ was negative (a net sink for OH) so that its contribution was also negative at night. The same amounts of radical loss or production from equilibrium reactions (e.g., $HO_2 + NO_2 \leftrightarrow HNO_4$; $CH_3COO_2 + NO_2 \leftrightarrow PAN$) was excluded from radical initiation or termination. $T(RO_x)_{NO_2+CH_3COO_2}$ represents the net PAN formation.

Line 65 – please cite Slater, ACP, 2020 (https://doi.org/10.5194/acp-20-14847-2020) and Whalley, ACP, 2021 (https://doi.org/10.5194/acp-21-2125-2021) as examples of the importance of HONO photolysis in OH formation.
Response: Improved as suggested.

Section 2.2.2 could do with more explanation/clarity, e.g. line 160, "significantly enhanced" by how much?
Response: In Sce-3 we enlarged $\gamma_a$ from $2\times10^{-5}$ to $1.2\times10^{-3}$ or EF from 7 to 400 (see Section 3.2.3) to test whether the aerosol-derived sources could well explain the observations.

Line 170 – was any back-trajectory analysis performed?
Response: We added back trajectory results in Figure S2. And the related text in Section 3.1.1: "Air masses observed at this site originated from multiple directions, including west, south, and east, which are shown in the wind rose plot in Figure S2A. Wind speed was generally low, with an average of about 2 m s$^{-1}$. In addition, the wind rose results generally agree well with HYSPLIT trajectory results in Figure S2B and S2C."

[Figure]

Figure S2: (A): Wind rose plot for the wind measurements at the foot of Mt. Tai; (B) and (C): 1-day back trajectory from HYSPLIT (https://www.ready.noaa.gov/HYSPLIT.php).

Line 194 and Table 3 – please cite Crilley, AMT, 2019 (https://doi.org/10.5194/amt-12-6449-2019) as a more recent example of HONO levels in Beijing.
Response: Improved as suggested.

Section 3.1.2/line 208 – misleading – the chemiluminescence method is fine (for the measurement of NO), it is the NO2 (to NO) converter chemistry that gives rise to interferences from other NOy

species.

Response: The reviewer is right. We improved the text as follows: "As the most important HONO precursor, accurate measurement of $NO_2$ plays a key role in analyzing HONO formation. The $NO_x$ monitor used in this study could specifically detect NO. To measure $NO_2$, a Molybdenum converter is used to convert $NO_2$ to NO. However, this chemical conversion process suffers from the interference of other $NO_y$ species (Villena et al., 2012)"

Lines 211, 214, and 215 – what is organic nitrates*? This is defined as RONO2 and ROONO2 in the Figure 3 caption, but please include the same definition in the main text.

Response: In the main text, we added a sentence "In this study, we defined the sum of $RONO_2$ and $ROONO_2$ as organic nitrates*" to make it consistent with Figure 3 caption.

Line 224 – the time is given as 11:00, should it really be 13:00?

Response: Yes, it should be 13:00. We corrected it.

Line 326 – the EF value of 15.6 appears to have come from your companion paper, therefore please reference this.

Response: Improved as suggested.

Line 433 – please give the average percentage contribution of HO2+NO – same for HONO photolysis on line 434.

Response: Average contributions of $HO_2$+NO (70%) and HONO photolysis (11%) were added in this sentence.

Line 446 – delete "and OH" – OH reactivity is not affected by [OH].

Response: Improved as suggested.

Line 501 – give reference for the summertime O3 increase in the NCP.

Response: Related references (Han et al., 2020; Li et al., 2019; Sun et al., 2016, 2019) are added.

Line 508 – ozonolysis chemistry also produces RO2

Response: Yes, $RO_2$ could also be produced during alkenes ozonolysis. This has already been considered in radical initiation (Figure 13). We improved the text: "Measurements on other radical precursors, such as $H_2O_2$ (through photolysis to produce OH), HCHO (through photolysis to produce $HO_2$), and alkenes (through ozonolysis via Criegee intermediate to produce OH, $HO_2$, and $RO_2$)."

**References**

Han, S., Yao, Q., Tie, X., Zhang, Y., Zhang, M., Li, P. and Cai, Z.: Analysis of surface and vertical measurements of $O_3$ and its chemical production in the NCP region, China, Atmos. Environ., 241, 117759, doi:10.1016/j.atmosenv.2020.117759, 2020.

Li, K., Jacob, D. J., Liao, H., Shen, L., Zhang, Q. and Bates, K. H.: Anthropogenic drivers of 2013–2017 trends in summer surface ozone in China, Proc. Natl. Acad. Sci. U. S. A., 116(2), 422–427, doi:10.1073/pnas.1812168116, 2019.

Sun, L., Xue, L., Wang, T., Gao, J., Ding, A., Cooper, O. R., Lin, M., Xu, P., Wang, Z., Wang, X., Wen, L., Zhu, Y., Chen, T., Yang, L., Wang, Y., Chen, J. and Wang, W.: Significant increase of summertime ozone at Mount Tai in Central Eastern China, Atmos. Chem. Phys., 16(16), 10637–10650, doi:10.5194/acp-16-10637-2016, 2016.

Sun, L., Xue, L., Wang, Y., Li, L., Lin, J., Ni, R., Yan, Y., Chen, L., Li, J., Zhang, Q. and Wang, W.: Impacts of meteorology and emissions on summertime surface ozone increases over central eastern China between 2003 and 2015, Atmos. Chem. Phys., 19(3), 1455–1469, doi:10.5194/acp-19-1455-2019, 2019.

Villena, G., Bejan, I., Kurtenbach, R., Wiesen, P. and Kleffmann, J.: Interferences of commercial $NO_2$ instruments in the urban atmosphere and in a smog chamber, Atmos. Meas. Tech., 5(1), 149–159, doi:10.5194/amt-5-149-2012, 2012.

---

## Author Response (AR2)

**Response to 2nd round review**

**1. Editor's comments to the author:**

Please address as quantitatively and in as much detail as possible the comments of the re-review from Reviewer #2.

These predominantly concern the quantitative implications of the selection of MLH and also the implications of the results for other locations.

Response: Thanks for the comments. We added our understanding and corresponding texts about MLH for HONO in the manuscript. Besides, a new section (Section 3.3.4) was supplemented to explain the translation of new methods we developed and conclusions we obtained in this study to further studies worldwide.

Please see the point-to-point response below (Comments in Black; Response in Blue; Changes in Red).

**2. Report #1 from Reviewer #2**

The manuscript has been improved after revision, and can be accepted to publish.

Response: Thank you.

**3. Report #2 from Reviewer #1**

ACP-2021-531 second review. The numbered comments correspond to the authors' replies to reviewer #1 in document "acp-2021-531-AC1-supplement" and line numbers to the authors' revised manuscript.

[Comment 1] This is a high quality observational dataset of HONO and related parameters that merits publication. My main concerns from the first review focused on the data interpretation side of the work, and particularly because the outcomes from the authors' 0-D modeling analysis rely heavily on the "correct" choice of the mixing layer height. MLH is set at 50 m, and two sensitivity tests are performed with the MLH set at 35 and 100 m.

The authors' response provides further qualitative discussion of the MLH based on their study and other literature studies. This is helpful. But it hasn't fully answered the question. I was hoping that the authors could be *quantitative* in their response and thus they would add numerical values of the MLH found in other studies into the main text, against which the reader can judge the authors' choice of the MLH. This would give confidence that 50 m is indeed a reasonable, *objective* choice. MLH = 50 m brings the authors' simple 0-D model into closest agreement with their HONO observations. But a different, more sophisticated model might produce a significantly different optimum MLH. Thus "50 m is best" is a consequence of the model, and not necessarily what is happening in the atmosphere itself. This part of the manuscript still needs further work:

* I accept the authors' response that it isn't possible to prove the MLH by comparing these ground-based observations with HONO measured at the mountain summit in their comparison paper.

* Were any numerical values of the MLH for HONO derived/quoted in the papers by Brown et al (2013) or Vandenboer et al (2013) or any other studies in the literature? If so, please add the MLH numbers into the main text.

* Numerical data of vertically resolved HONO measurements from Xing et al (2021) were given in

the main text (and repeated in the authors' response). Did Xing et al derive a number for the MLH? Can their data be fitted to provide a value(s) of the MLH?

* Do the sensitivity tests performed at 35 m and 100 m cover the likely range of how the MLH varies with wind speed, time of day, day vs night etc? i.e. do 35 and 100 m encompass the lower and upper limits of what is happening at their measurement site?

I'm entirely comfortable with the authors' conclusion that ground-based HONO sources dominate at the Tai'an city measurement site, and consequently the MLH for HONO is small (of the order of 50 m). I also recognise that the exact value of the MLH is changeable with conditions at their measurement site, and the MLH will likely be different at other locations. But at the moment the manuscript asks its readers to accept 50 m as the correct choice based on the authors' assertion. And this has consequences for the understanding of the atmospheric chemistry that follows.

Response to the above comments concerning MLH:

Thanks for the further comments. A proper choice of MLH is challenging but very important for 0D box model outcomes when constrained with near ground surface measurements, which is the reason why the reviewer and we are very serious with this issue.

In the supporting information, we added a new section to explain in detail.

**2. A detailed explanation for the used MLH**

Currently, some studies with ground measurements directly used the boundary layer height (BLH, 1-2 km at noon) instead of MLH. This would largely underestimate the contribution of ground-derived sources, leading to the misunderstanding of HONO formation. In the present study, we could not conclude that the MLH of 50 m (and sensitivity tests for 35-100 m) is the best, but it significantly reduces the uncertainties compared to the use of BLH. Additionally, a reasonable MLH for model study on ground HONO measurements should be in the range we tested. See the explanation below.

Here we assume that the ground surface is the main source of HONO in the atmosphere. This is for example confirmed by recent MAX-DOAS studies (Garcia-Nieto et al., 2018; Ryan et al., 2018; Wang et al., 2019; Xing et al., 2021), in which strong gradients were observed in the lower daytime atmosphere. The gradients can be explained by fast photolysis of HONO during the vertical updraft from the ground surfaces (source region of HONO) during the daytime. The mixing layer higher (MLH), i.e., the height to which ground surface produced HONO will be transported, will depend on both the photolytic lifetime of HONO (inverse of J(HONO)) and the vertical mixing of the atmosphere described e.g., by the eddy-diffusion coefficient. In response to the solar zenith angle (SZA), the lifetime of HONO will decrease from morning to noon, which will solely lead to a decreasing MLH. In contrast, caused by the increasing turbulence from morning to noon (Jacob, 1999), the vertical transport of HONO increase (increasing MLH). If the vertical transport is increasing in the same way as the photolytic lifetime of HONO is decreasing, both effects will exactly compensate leading to a constant MLH as used for simplicity in the present study.

Generally, the MLH could be defined as the height where the HONO concentration – or more precisely the excess HONO concentration exceeding (height-dependent) HONO$_{PSS}$ – has decreased to 1/e from its ground surface concentration. Caused by the gradients, a formal source determined from the excess over PSS will be height dependent and stronger when measured close to the ground as done in the present study. The reason for this problem is that the sources are not correctly implemented as flux from the ground surface (molecules $m^{-2}$ $s^{-1}$) in a 1D vertical model, but are mathematically treated as a gas phase source in a homogeneously mixing box model, which we used here for simplicity. Thus, the box height has to be even lower than the above-defined MLH and will

be better described by the height where the HONO mixing ratio is decreasing to lower values than the measured near the ground surface. A better definition of the height used would be the homogeneous mixing height of the 0D box, for which we used the term MLH for simplicity.

Then we did several steps to scale the MLH used in this study.

A minimum MLH of 35 m was derived based on the assumption that all the $P_{unknown}$ could be wholly explained by photosensitized heterogeneous $NO_2$ reaction on the ground surface in our recent study (Xue et al., 2021).

To scale the maximum of the MLH of HONO, theoretically, the vertical turbulence process within the lifetime of HONO should be considered. For instance, Zhang et al. (2009) estimated the maximum vertical transport distance by turbulent diffusion (Jacob, 1999). A maximum of 350 m at noontime that HONO could reach was obtained. Therefore, MLH for HONO should be much lower than 350 m, which is in agreement with vertical measurements.

Brown et al. (2013) and Vandenboer et al. (2013) are under the same project of Nitrogen, Aerosol Composition, and Halogens on a Tall Tower (NACHTT-11) and the latter one was focused on HONO formation. Vandenboer et al. (2013) conducted similar model simulations with a model height of 150 m. They found significant underestimation in HONO levels, which was attributed to the higher model height compared to the measurement height of 20 m. Hence, to model measurements near the ground surface, a lower MLH than 150 m is needed.

Vertical measurements could furtherly scale the MLH. A declining HONO trend with altitude was frequently observed in previous vertical measurements (Kleffmann et al., 2003; Meng et al., 2020; Vandenboer et al., 2013; Vogel et al., 2003; Xing et al., 2021; Ye et al., 2018; Zhang et al., 2009). We would like to take the measurements in Germany (Vogel et al., 2003), the USA (Vandenboer et al., 2013) and China (Xing et al., 2021) as examples to scale the MLH. From the ground level (4-10 m) to 100 m above the ground surface, Vogel et al. (2003), Vandenboer et al. (2013), and Xing et al. (2021) observed declining HONO levels from ~0.6 to 0.3 (a representative case from Figure 4), from 0.6 to 0.3 (case from Figure 8), and from 4.8 to 1.6 ppbv (case from Figure 5), respectively. All of those cases suggest that near-ground surface measurements were more weighted by ground-derived sources. Moreover, this phenomenon was observed during their whole campaigns including daytime and nighttime, suggesting a similar level of MLH. Hence, the maximum of MLH could be furtherly scaled to 100 m for near-ground surface measurements.

In summary, 0-D modeling with the utilization of ~50 m level could represent a general MLH for studying HONO measurements near the ground surface. Nevertheless, we still should highlight that accompanied efforts, e.g., performing sensitivity tests, should always be made to underline the uncertainties.

Regrading deriving MLH from vertical measurements like Xing et al. (2021), we need to conduct 1D modeling simulations with reasonable transport and a real surface flux of HONO. The model results should be compared with near-ground surface measurements or gradient measurements. However, currently, we don't have the tool of a 1D model and gradient measurements. Instead, in this study, we tried to scale the MLH using the above methods, which significantly improved the model performance.

In Section 3.2.2.4 of the main text, we added a brief discussion on how we scale MLH:

A similar phenomenon could also be found in tower-based vertical measurements in Germany and USA. For instance, from the ground level (4-10 m) to 100 m above the ground surface, Vogel et al.

(2003) and Vandenboer et al. (2013) observed similarly declining HONO levels from ~0.6 to 0.3 (representative cases from Figure 4 in Vogel et al. (2003) and Figure 8 in Vandenboer et al. (2013)), respectively. All of those cases suggest that near-ground surface measurements were more weighted by ground-derived sources. Moreover, this phenomenon was observed during their whole campaigns including daytime and nighttime, suggesting a similar level of MLH. Hence, the maximum of MLH could be furtherly scaled to 100 m for near-ground surface measurements. A minimum MLH of 35 m was derived based on the assumption that all the $P_{unknown}$ could be wholly explained by photosensitized heterogeneous $NO_2$ reaction on the ground surface in our recent study (Xue et al., 2021). Therefore, in the present study, the MLH for HONO was set at a constant height of 50 m, with sensitivity tests performed with the MLH set at 35 and 100 m. In general, ~50 m level could represent a general MLH for 0D models to study HONO measurements near the ground surface. Similar values (25 – 100 m) were also used in previous box model studies (Harrison et al., 1996; Lee et al., 2016; Xue et al., 2020, 2021). Nevertheless, it should be highlighted that a box model as used in the present study is not an ideal tool for studying a ground source when comparing with near-ground surface measurements in the atmosphere. For the future, gradient measurements are recommended, which should be compared with 1-D model simulations.

[Comment 2]
Fig 4 & line 269. The addition of the J-HONO photolysis frequency to the diurnal profile in Fig 4 is informative and welcome. Likewise the more explicit reference in the text to the asymmetric shape of P_unknown. I think Line 269 should now read "Note that the profile of P_unknown *is* asymmetric around *12:00 solar noon*, indicating the unknown source is not simply photolytic…" The important point here is that the diurnal profile exhibited by P_unknown is offset with respect to the peak photolysis activity of J-HONO and/or J-NO2 at solar noon (rather than any asymmetry before/after the peak in P_unknown at 11:00).
Response: Changed as the reviewer suggested.

It is good the authors have added campaign-averaged diurnal profiles of HONO, O3, NO, NO2 etc as a new figure S3 in the supporting information (also requested by referee #3). Personally I would put this new figure into the main body of the paper because its information is very important for telling the scientific story.
Response: Thanks for the suggestion. The average diurnal profiles were moved to the main body.

Continuing discussion of the sentence that began on line 269: The second half of the sentence reads "…but also includes its precursors that also have an asymmetric distribution (e.g., NO2, Figure S3)". The text need to be much more explicit here. Panel (E) in new figure S3 certainly shows that NO2 concentrations were higher in the morning than in the afternoon – so are the authors saying the faster P_unknown HONO production rate observed before midday (and peaking at 11:00) is attributed to the greater NO2 concentrations present in the morning? This would fit with the authors' finding that P_unknown has its biggest contribution from heterogeneous NO2 to HONO conversion at the ground (Figure 8).
Response: Yes, this is a good point. We added
Note that the profile of $P_{unknown}$ is asymmetric around 12:00 solar noon, indicating the unknown source is not simply photolytic but also includes its precursors. For instance, higher $NO_2$ levels were

observed in the morning than that in the afternoon (Figure 3E), which preliminarily implies the importance of $NO_2$-to-HONO conversion.

[Comment 3]

I asked the authors to expand about how the conclusions of this paper translate to other locations in China and other countries? The revisions here are disappointing and I had hoped to see more discussion. The authors have added a sentence at line 523 "Model results may have uncertainties but shed light on the atmospheric chemistry in this polluted region", which is certainly true but fails to answer the reviewer's question.

Response: Regarding implications of conclusions in this study to further studies worldwide, we added a new section as follows:

3.4 Implications for Further Studies

For the first time, we considered HONO and $NO_x$ lifetimes to quantify the contribution of direct emission to daytime HONO formation. The method developed here remarkably reduced the overestimation of contribution from direct emission. It is universal and should be used for all ground measurements to quantify the contribution of direct emission to daytime HONO formation.

In the present study, we also conclude that heterogeneous $NO_2$ reaction on the ground surfaces is the major HONO source. Constraints on aerosol-derived sources, including $NO_2$ uptake on the aerosol surfaces and particulate nitrate photolysis, are conducted by our measurements at the summit of Mt. Tai and recent laboratory studies. Therefore, it could be expected that similar conclusions can be found in other studies when considering ground measurements. Additionally, parametrizations of HONO sources used for box model simulations are applicable for other studies. The values of some parameters, such as $NO_2$ uptake coefficients, MLH and particulate nitrate photolysis frequency, are obtained or derived from laboratory or field studies. Further studies may improve the understanding of the variation of those parameters, for instance, $NO_2$ uptake coefficients may vary with locations that have different landscapes. In particular, based on three vertical measurements in Germany, the USA, and China, similar levels of MLH (<100 m) were derived, indicating the potential application of this method to ground measurements worldwide. This could significantly reduce the underestimation of HONO formation from ground-derived sources compared to models in which the BLH was used. Meanwhile, sensitivity tests should be conducted and uncertainties should be discussed accordingly.

It has been recognized that HONO photolysis could initiate daytime atmospheric chemistry in the early morning and also acts as a substantial OH source during the daytime in polluted regions. The significant contribution and impact of HONO on radical levels could motivate studies by chemistry-transport models, most of which have currently not included HONO chemistry.

Furthermore, $O_3$ pollution is becoming a key environmental issue in China. While high levels of $O_3$ are present, moderate levels of $NO_x$ are frequently accompanied. In this case, $NO_3$ chemistry could make a considerable contribution to atmospheric oxidization capacity, especially during the nighttime. Its follow-up impacts on atmospheric composition, such as the formation of nitrate and SOA, need further field measurements and model quantifications.

General comments:

The authors have done a lot of work on their manuscript. Overall, the revised manuscript is a great improvement because the authors have acted on the comments from 3 detailed, extensive reviews.

The authors have also thoroughly proof-read their manuscript, which has removed most of the English language problems in the original manuscript. Whilst the authors thanked the reviewers in the authors' individual responses to the 3 referees, perhaps they might also consider acknowledging the referees' efforts in the acknowledgements section of the main manuscript?

Response: Yes, we should do that. In the acknowledge section, we added the below sentence:

We appreciate the three anonymous reviewers and the editor, John Orlando, for their careful reading of our manuscript and many insightful comments, suggestions, and discussion.

Minor errors (These are errors I spotted whilst reviewing, so not an exhaustive list):

Caption to Fig 4, line 260. "The relative contribution of NO + OH and … *are* shown…in the pie charts". I also suggest moving this sentence to the end of the caption, i.e. to after the sentence about the "blue shaded area" which seems to refer to the main plot (and not the blue shading in the pie charts). Or does the blue shading in the plot and the pie charts both represent the same observed-minus-model differences? I was unsure, please clarify and re-word as necessary.

Response: The caption was improved as:

Simulated HONO by the default mechanism (Sce-1, left axis) compared with the observations (Obs, left axis), unknown source strength ($P_{unknown}$, right axis), HONO photolysis frequency (J(HONO), right axis). The shaded area in blue in the plot and the pie charts represent the difference between the observation and modeled values. The relative contributions of NO + OH to the observations at night (19:00 – 4:50) and day (5:00 – 18:50) are shown in the left and the right pie charts, respectively.

Line 267. "… were calculated *from* the measurements [plural]…".

Response: Changed as commented.

Line 317. "…upper limit derived from the summit measurements [plural] *in our companion paper*, see Xue et al. (2021b)…"

Response: Changed as commented.

Line 389. "The only major exception was a period of heavy rain from 25th to 28th June…"

Response: Changed as commented.

Line 392. "While the modified model could generally predict…, [add comma & delete "but"] it largely failed…"

Response: Corrected as commented.

Fig 10, line 470. The authors have improved the colors and now the distinctions are a lot clearer between the various OH sources in panel (A) and NO3 sources in panel (B). However the new fig 10 now uses a light scale of pastel colors. I actually preferred the style of bold colors in panels (C) and (D) of the original – it was easier to distinguish the bold colors.

Response: The figure was modified by using style of bold colors.

[Figure]

**Figure 11: Production rates (P), loss rates (L) and reactivities of radicals. (A): L and P of OH; (B): L and P of NO₃; (C): Reactivities of OH and (D): Reactivities of NO₃. In (A) and (B), the top-3 sources or sinks are shown, and all the others are summarized in "Other sources" or "Other sinks". In (C), OH reactivities with different families of the measured species are shown and reactivities with all the unmeasured species are summarized in "Others". In (D), NO₃ reactivities from top-3 reactions are shown and all the others are summarized in "Others".**

**References**

Brown, S. S., Thornton, J. A., Keene, W. C., Pszenny, A. A. P., Sive, B. C., Dubé, W. P., Wagner, N. L., Young, C. J., Riedel, T. P., Roberts, J. M., Vandenboer, T. C., Bahreini, R., Öztürk, F., Middlebrook, A. M., Kim, S., Hübler, G. and Wolfe, D. E.: Nitrogen, Aerosol Composition, and Halogens on a Tall Tower (NACHTT): Overview of a wintertime air chemistry field study in the front range urban corridor of Colorado, J. Geophys. Res. Atmos., 118(14), 8067–8085, doi:10.1002/jgrd.50537, 2013.

Garcia-Nieto, D., Benavent, N. and Saiz-Lopez, A.: Measurements of atmospheric HONO vertical distribution and temporal evolution in Madrid (Spain) using the MAX-DOAS technique, Sci. Total Environ., 643, 957–966, doi:10.1016/j.scitotenv.2018.06.180, 2018.

Harrison, R. M., Peak, J. D. and Collins, G. M.: Tropospheric cycle of nitrous acid, J. Geophys. Res. Atmos., 101(D9), 14429–14439, doi:10.1029/96JD00341, 1996.

Jacob, D. J.: Introduction to atmospheric chemistry, Princeton University Press., 1999.

Kleffmann, J., Kurtenbach, R., Lörzer, J., Wiesen, P., Kalthoff, N., Vogel, B. and Vogel, H.: Measured and simulated vertical profiles of nitrous acid - Part I: Field measurements, Atmos. Environ., 37(21), 2949–2955, doi:10.1016/S1352-2310(03)00242-5, 2003.

Lee, J. D., Whalley, L. K., Heard, D. E., Stone, D., Dunmore, R. E., Hamilton, J. F., Young, D. E., Allan, J. D., Laufs, S. and Kleffmann, J.: Detailed budget analysis of HONO in central London reveals a missing daytime source, Atmos. Chem. Phys., 16(5), 2747–2764, doi:10.5194/acp-16-2747-2016, 2016.

Meng, F., Qin, M., Tang, K., Duan, J., Fang, W., Liang, S., Ye, K., Xie, P., Sun, Y., Xie, C., Ye, C., Fu, P., Liu, J. and Liu, W.: High-resolution vertical distribution and sources of HONO and NO₂ in the nocturnal boundary layer in urban Beijing, China, Atmos. Chem. Phys., 20(8), 5071–5092, doi:10.5194/acp-20-5071-2020, 2020.

Ryan, R. G., Rhodes, S., Tully, M., Wilson, S., Jones, N., Frieß, U. and Schofield, R.: Daytime HONO, NO₂ and aerosol distributions from MAX-DOAS observations in Melbourne, Atmos. Chem. Phys.

Discuss., (2), 1–27, doi:10.5194/acp-2018-409, 2018.

Vandenboer, T. C., Brown, S. S., Murphy, J. G., Keene, W. C., Young, C. J., Pszenny, A. A. P., Kim, S., Warneke, C., De Gouw, J. A., Maben, J. R., Wagner, N. L., Riedel, T. P., Thornton, J. A., Wolfe, D. E., Dubé, W. P., Öztürk, F., Brock, C. A., Grossberg, N., Lefer, B., Lerner, B., Middlebrook, A. M. and Roberts, J. M.: Understanding the role of the ground surface in HONO vertical structure: High resolution vertical profiles during NACHTT-11, J. Geophys. Res. Atmos., 118(17), 10,155-10,171, doi:10.1002/jgrd.50721, 2013.

Vogel, B., Vogel, H., Kleffmann, J. and Kurtenbach, R.: Measured and simulated vertical profiles of nitrous acid - Part II. Model simulations and indications for a photolytic source, Atmos. Environ., 37(21), 2957–2966, doi:10.1016/S1352-2310(03)00243-7, 2003.

Wang, Y., Dörner, S., Donner, S., Böhnke, S., De Smedt, I., Dickerson, R. R., Dong, Z., He, H., Li, Z., Li, Z., Li, D., Liu, D., Ren, X., Theys, N., Wang, Y., Wang, Z., Xu, H., Xu, J. and Wagner, T.: Vertical profiles of NO$_2$, SO$_2$, HONO, HCHO, CHOCHO and aerosols derived from MAX-DOAS measurements at a rural site in the central western North China Plain and their relation to emission sources and effects of regional transport, Atmos. Chem. Phys., 19(8), 5417–5449, doi:10.5194/acp-19-5417-2019, 2019.

Xing, C., Liu, C., Hu, Q., Fu, Q., Wang, S., Lin, H., Zhu, Y., Wang, S., Wang, W., Javed, Z., Ji, X. and Liu, J.: Vertical distributions of wintertime atmospheric nitrogenous compounds and the corresponding OH radicals production in Leshan, southwest China, J. Environ. Sci., 105, 44–55, doi:10.1016/j.jes.2020.11.019, 2021.

Xue, C., Zhang, C., Ye, C., Liu, P., Catoire, V., Krysztofiak, G., Chen, H., Ren, Y., Zhao, X., Wang, J., Zhang, F., Zhang, C., Zhang, J., An, J., Wang, T., Chen, J., Kleffmann, J., Mellouki, A. and Mu, Y.: HONO Budget and Its Role in Nitrate Formation in the Rural North China Plain, Environ. Sci. Technol., 54(18), 11048–11057, doi:10.1021/acs.est.0c01832, 2020.

Xue, C., Ye, C., Zhang, C., Catoire, V., Liu, P., Gu, R., Zhang, J., Ma, Z., Zhao, X., Zhang, W., Ren, Y., Krysztofiak, G., Tong, S., Xue, L., An, J., Ge, M., Mellouki, A. and Mu, Y.: Evidence for Strong HONO Emission from Fertilized Agricultural Fields and its Remarkable Impact on Regional O 3 Pollution in the Summer North China Plain, ACS Earth Sp. Chem., 5(2), 340–347, doi:10.1021/acsearthspacechem.0c00314, 2021.

Ye, C., Zhou, X., Pu, D., Stutz, J., Festa, J., Spolaor, M., Tsai, C., Cantrell, C., Mauldin III, R. L., Weinheimer, A., Hornbrook, R. S., Apel, E. C., Guenther, A., Kaser, L., Yuan, B., Karl, T., Haggerty, J., Hall, S., Ullmann, K., Smith, J. and Ortega, J.: Tropospheric HONO distribution and chemistry in the southeastern US, Atmos. Chem. Phys., 18(12), 9107–9120, doi:10.5194/acp-18-9107-2018, 2018.

Zhang, N., Zhou, X., Shepson, P. B., Gao, H., Alaghmand, M. and Stirm, B.: Aircraft measurement of HONO vertical profiles over a forested region, Geophys. Res. Lett., 36(15), L15820, doi:10.1029/2009GL038999, 2009.

---

## Author Response (AR3)

**Response to Editor Comments**

**Comments to the author**:

I think the paper is essentially ready to be published, but I do have some technical (mostly grammatical) suggestions to offer. Please see the non-public comments for a list of these suggestions.

Thank you for your attention to these details.

Non-public comments to the Author:

The paper is very easily understandable, and I appreciate the authors' efforts to respond to reviewers' comments and to improve the paper. However, I do have a few suggestions to further improve the grammar of the paper. Please replace the current wording with the words/phrases listed below.

Response: Many thanks for the suggestions and corrections. And thanks for the management of our manuscript. Please see the response below.

Supplementary Line 69: Maybe add one sentence of introduction here to reiterate that the choice of MLH is important.

Response: We added a sentence: A proper level of the employed MLH is of significant importance for parameterizing ground-derived HONO sources, as discussed in Section 3.2.2.4 of the main text.

All below are improved as the editor suggested. Thanks again.

Line 64: quantification, (instead of quantifications)

Line 77: Change 'Besides' to 'In addition'

Line 79: Replace 'concerning' with 'due to'

Line 85: Change 'organic nitrates' to 'organic nitrate'

Line 99-100: I suggest the following: '…or loss (e.g., to oxidize primary pollutants) in the high-O3 region of the NCP (Lu et al., …)

Line 136: Replace 'auto' with 'automated'

Line 159: Replace 'with' with 'that involved'.

Line 164: "…was reduced by a factor of 10, and aerosol-derived sources…"

Line 171-172: I suggest "except for slight rain (<10 mm) on 9th, 10th, 13th, and 28th and heavy rain (≈100mm) at night …"

Line 192: I suggest "… some fresh plumes that contained higher NO concentrations."

Line 273: "were calculated from the measurements…"

Line 277: Delete 'that'

Line 347: "a value of 7 was reported from a recent field study…"

Line 386: Maybe "… maximum of MLH could be reasonably assumed to be 100m …"

Line 398: The word 'Considering' can be deleted, I think.

Line 439: "is still uncertain". I think that is what is meant here?

ecomposition' instead of "deposition".

Line 510: "rates"
Line 595: I suggest "These high levels of O3 are often accompanied by moderate levels of NOx."

Supplement:
Line 68: "…for the MLH employed"
Line 81-82: "the vertical transport of HONO will increase…"
Line 100: "Brown et al. (2013) and Vandenboer et al. (2013) both resulted from the same project, Nitrogen, …"
Line 105: "Vertical measuremnts can further constrain the MLH."
Line 113: I suggest something like this, if this captures the meaning of what you are trying to say: "Hence, a maximum MLH of 100 m appears appropriate for interpretation of near-ground surface measurements."